# PARTITION GENERATIVE MODELING: MASKED MODELING WITHOUT MASKS

**Justin Deschenaux**[1]*  **Lan Tran**[1]  **Caglar Gulcehre**[1]

[1]EPFL, Lausanne, Switzerland

## ABSTRACT

Masked generative models (MGMs) can generate tokens in parallel and in any order, unlike autoregressive models (ARMs), which decode one token at a time, left-to-right. However, MGMs process the full-length sequence at every sampling step, including [MASK] tokens that carry no information. In contrast, ARMs process only the previously generated tokens. We introduce "Partition Generative Models" (PGMs), which replace masking with partitioning. Tokens are split into two groups that cannot attend to each other, and the model learns to predict each group conditioned on the other, eliminating [MASK] tokens entirely. Because the groups do not interact, PGMs can process only the clean tokens during sampling, like ARMs, while retaining parallel, any-order generation, like MGMs. On Open-WebText, PGMs achieve 5–5.5× higher throughput than MDLM while producing samples with lower Generative Perplexity. On ImageNet, PGMs reach comparable FID to MaskGIT with a 7.5× throughput improvement. With twice as many steps, the FID improves to 4.56 while remaining 3.9× faster than MGMs. Finally, PGMs remain compatible with existing MGM samplers and distillation methods.

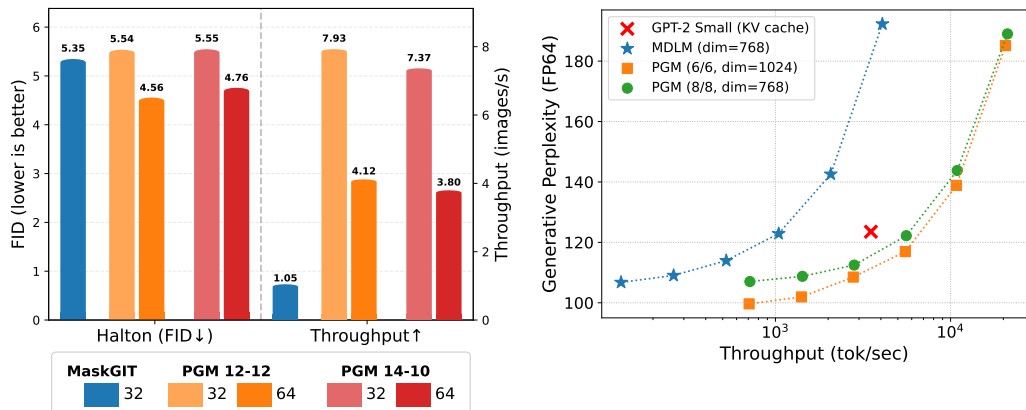

Figure 1: **(Left)**: On ImageNet, using the Halton sampler, PGM (ours), reaches similar FID as MaskGIT with a 7.5× speedup. By sampling with twice as many steps, PGM reaches an FID of 4.56 while remaining 3.9× faster. **(Right)**: On OpenWebText, PGM achieves a better Generative Perplexity with a 5.3× higher sampling throughput compared to MDLM (an MGM for text), at a context length of 1024.

## 1 INTRODUCTION

Masked generative models (MGMs) offer two key advantages over autoregressive models (ARMs): they can generate tokens in parallel and in any order, rather than one-by-one, left-to-right. These properties have led to strong results across images (Chang et al., 2022), video (Yu et al., 2023; Villegas et al., 2022), audio (Comunità et al., 2024), and language (Austin et al., 2023; Lou et al., 2024; Sahoo et al., 2024; Shi et al., 2025; Campbell et al., 2024; Gat et al., 2024). However, MGMs

---

*Correspondence to justin.deschenaux@epfl.ch.

are slow at inference. Indeed, at every sampling step, they process the full-length sequence, including [MASK] tokens that carry no information, whereas ARMs process only the previously generated tokens. This limits the practicality of MGMs in large-scale and real-time settings, and is a crucial disadvantage for test-time compute scaling (Snell et al., 2024; Wu et al., 2024) compared to ARMs.

Addressing the inference inefficiency of MGMs is not trivial because training and sampling must be consistent. MGMs are trained with bidirectional architectures over the full sequence, so every hidden representation depends on all $L$ positions, including masked ones. Furthermore, naively decoding tokens block-by-block means feeding the model shorter sequences at inference, which differs from training and leads to poor sample quality (Deschenaux & Gulcehre, 2024).

Prior work addresses the slow inference of MGMs from different angles. Decoding more tokens per step increases throughput, but degrades sample quality. Distillation (Deschenaux & Gulcehre, 2025; Zhu et al., 2025; Sahoo et al., 2025a) reduces the number of sampling steps, but each step remains equally expensive, and distillation can affect the sample diversity (Gandikota & Bau, 2025). Block Diffusion (Arriola et al., 2025) enables partial KV-caching by generating tokens block-by-block, but sacrifices the any-order generation capability. None of these approaches make *individual sampling steps cheaper* while preserving the full flexibility of MGMs.

We introduce *Partition Generative Models* (PGMs), which replace masking with partitioning. Tokens are split into two disjoint groups, and a group-wise attention mechanism ensures that no information flows between them. The model learns to predict each group conditioned on the other, eliminating [MASK] tokens entirely. Because the two groups do not interact, PGMs process only the clean tokens during sampling, just like ARMs, while retaining the ability to generate tokens in parallel and in any order, like MGMs. We propose the *Partition Transformer*, a dedicated architecture that prevents information flow between groups.

**Contributions** (1) We introduce PGMs and the Partition Transformer, a new architecture that enables MGM-style parallel, any-order generation without [MASK] tokens. PGMs are compatible with existing MGM samplers (Besnier et al., 2025) and distillation methods (Deschenaux & Gulcehre, 2025), making them a drop-in replacement. (2) On OpenWebText (Gokaslan & Cohen, 2019), PGMs generate samples with lower Generative Perplexity than MDLM (Sahoo et al., 2024) and reach similar downstream task performance, before and after distillation, while **achieving 5–5.5× higher throughput**. On ImageNet, PGMs reach comparable FID to MaskGIT (Chang et al., 2022) with a **7.5× throughput improvement**. (3) We show that PGM are trained with denser supervision than MGM. Since each group predicts the other, a single sequence yields two complementary training signals. This reduces gradient variance and yields a **1.95 reduction in validation perplexity** on LM1B (Chelba et al., 2014) compared to MDLM with the same number of layers.

## 2 BACKGROUND

### 2.1 SEQUENCE MODELING

We consider the task of generating sequences $\mathbf{x} = (x_1, \ldots, x_L)$ of length $L$ over a vocabulary $\mathcal{V} = \{0, \ldots, N-1\}$. The training dataset $\mathcal{D}$ contains finitely many sequences drawn from an unknown data distribution $p_{\text{data}}$ over $\mathcal{V}^L$. **Autoregressive models** (ARMs) factorize the distribution as $p_\theta(\mathbf{x}) = \prod_{i=1}^{L} p_\theta(\mathbf{x}_i \mid \mathbf{x}_{<i})$, where $\mathbf{x}_{<i}$ denotes the prefix before position $i$. Tokens are sampled sequentially, and because each conditional only depends on the prefix $\mathbf{x}_{<i}$, ARMs process only the previously generated tokens instead of the whole sequence.

### 2.2 MASKED GENERATIVE MODELS

MGMs augment the vocabulary with a special [MASK] token, absent from the training data. Given $\mathbf{x} \in \mathcal{D}$, let $\mathbf{z}_t$ denote a corrupted sequence where the token $\mathbf{z}_t^\ell$ at position $\ell$ is [MASK] with probability $p_t$, or the clean value $\mathbf{x}^\ell$ otherwise. The masking probability $p_t$ is an increasing function of $t \in [0, 1]$ with $p_0 = 0$ and $p_1 = 1$. MGMs train a denoiser $\mathbf{x}_\theta : \mathcal{V}^L \to \mathbb{R}^{L \times N}$ whose sampling distribution is modeled as factorized marginals: $\prod_{\ell:\mathbf{z}_t^\ell = [\text{MASK}]} p_\theta^\ell(. \mid \mathbf{z}_t)$. Because MGMs sample independently from each $p_\theta^\ell(. \mid \mathbf{z}_t)$, they cannot model arbitrary joint distributions, unlike ARMs. However, they

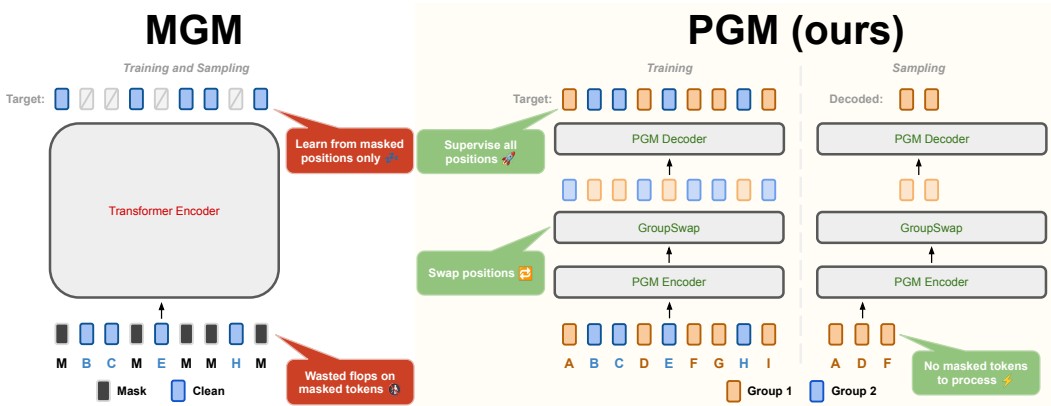

Figure 2: **Masked Generative Modeling (MGM) vs. Partition Generative Modeling (PGM). Left:** MGMs learn at masked positions only, and must process [MASK] tokens at inference. **Right:** PGMs partition tokens in two groups, learn from *all* positions and process clean tokens only during sampling. PGMs achieve >5.3x higher throughput on OpenWebText (context length 1024, 128 sampling steps).

can decode tokens in parallel. The training objective for MGMs is:

$$\mathcal{L}_{\text{MGM}} := \mathbb{E}_{\mathbf{x}\sim\mathcal{D}, t\sim\mathcal{U}[0,1]} \left[ w(t)\text{CE}(\mathbf{x}_\theta(\mathbf{z}_t; t), \mathbf{x}) \right], \tag{1}$$

where $w : [0,1] \to \mathbb{R}_{\geq 0}$ is a weighting function and $\text{CE}(\hat{\mathbf{x}}, \mathbf{x})$ denotes the cross-entropy loss over masked positions. To generate samples, MGMs start from a fully masked sequence and iteratively unmask subsets of positions over multiple evaluations of $\mathbf{x}_\theta$, selecting positions at random or based on confidence scores. We now describe two instantiations used in this work.

**MDLM**  Masked Diffusion Language Models (MDLM; Sahoo et al. (2024); Ou et al. (2025); Shi et al. (2025)) are MGMs for language modeling. Analogously to continuous diffusion (Sohl-Dickstein et al., 2015; Song & Ermon, 2020; Ho et al., 2020; Kingma et al., 2023), MDLM defines a forward process that corrupts clean data and a generative process that recovers samples from noise. The forward process is:

$$q_t(.|\mathbf{x}) := \text{Cat}(.; \alpha_t\mathbf{x} + (1-\alpha_t)\boldsymbol{\pi}), \tag{2}$$

where $\mathbf{x}$ is the one-hot representation, $\boldsymbol{\pi} = \mathbf{m}$ is the one-hot encoding of [MASK], and $\alpha_t$ is a strictly decreasing noise schedule with $\alpha_0 = 1, \alpha_1 = 0$. (2) is applied independently at every position. The posterior distribution is:

$$p_{s|t}(.|\mathbf{z}_t, \mathbf{x}) = \begin{cases} \text{Cat}(.; \mathbf{z}_t), & \mathbf{z}_t \neq \mathbf{m}, \\ \text{Cat}\left(.; \frac{(1-\alpha_s)\mathbf{m}+(\alpha_s-\alpha_t)\mathbf{x}}{(1-\alpha_t)}\right), & \mathbf{z}_t = \mathbf{m}. \end{cases} \tag{3}$$

To generate samples, we fix a decreasing sequence of times $1 = \tau_T > \cdots > \tau_0 = 0$, set $\mathbf{z}_{\tau_T}$ to the all [MASK] tokens sequence, and iteratively sample from

$$\mathbf{z}_{\tau_{i-1}} \sim p_{\theta, \tau_{i-1}|\tau_i}(. \mid \mathbf{z}_{\tau_i}) = p_{\tau_{i-1}|\tau_i}(.|\mathbf{z}_t, \mathbf{x}_\theta(\mathbf{z}_{\tau_i}, \tau_i)). \tag{4}$$

MDLM optimizes a variational bound on log-likelihood that reduces to (1) with $w(t) = \frac{\alpha_t'}{1-\alpha_t}$. Only masked positions contribute to the loss.

**MaskGIT**  MaskGIT (Chang et al., 2022) is an MGM that operates in the latent space of a pre-trained VQGAN (Esser et al., 2021) tokenizer. MaskGIT proposes tokens $\tilde{\mathbf{x}}^\ell$ at masked position and uses the predicted likelihood of the sampled token $\tilde{\mathbf{x}}^\ell$ as a confidence score: $c^\ell = \mathbf{x}_\theta^\ell(\mathbf{z}_t; t)_{\tilde{\mathbf{x}}^\ell}$. A predefined schedule determines the number of positions to unmask, and the most confident positions are kept. Tokens generated in earlier steps are kept unchanged. This differs from MDLM, which denoises at random positions. Besnier et al. (2025) observed that confidence-based sampling tends to decode spatially clustered tokens, since the denoiser is most confident near previously generated positions. Because MGMs sample independently from a product of marginals $\prod_{\ell\in\mathcal{S}} p_\theta(\mathbf{x}^\ell \mid \mathbf{z}_\tau)$ rather than the joint token distribution, decoding nearby tokens increases the risk of generating

inconsistent samples. By sampling according to a low-discrepancy sequence (Halton, 1964), Besnier et al. (2025) enforce a more uniform coverage of the space, which improves the FID and IS compared to confidence sampling.

**Classifier-Free Guidance**  Let $p_\theta(\mathbf{x} \mid c)$ denote a class-conditional distribution learned by an MGM, where $c \in \{0, \ldots, C-1\}$ is a class label and $p_\theta(\mathbf{x} \mid \varnothing)$ denotes the class-unconditional distribution. Let $\omega \geq 0$ control the guidance strength. Classifier-Free Guidance (CFG; Ho & Salimans (2022); Chang et al. (2022)) steers generation toward class $c$ by replacing $\log p_\theta(\mathbf{x} \mid c)$ during sampling with

$$\log \tilde{p}_\theta(\mathbf{x} \mid c) = (1 + \omega) \log p_\theta(\mathbf{x} \mid c) - \omega \log p_\theta(\mathbf{x} \mid \varnothing), \tag{5}$$

**Self-Distillation Through Time**  Self-Distillation Through Time (SDTT; Deschenaux & Gulcehre (2025)) accelerates sampling from MGMs by distilling a teacher trained for denoising with many steps into a few-steps student. Let $p_\theta^{(m)}$ denote the distribution of samples generated with $m$ steps using a denoiser $\mathbf{x}_\theta$, and let $p_\nu^{(k)}$ denote the distribution when using $k < m$ steps with a student denoiser $\mathbf{x}_\nu$. SDTT trains $\mathbf{x}_\nu$ with the following objective:

$$\min_\nu \ \mathbb{E}_{\mathbf{z}_0 \sim \mathcal{D}, \mathbf{z}_t \sim q_t(\mathbf{z}_t \mid \mathbf{z}_0)} \left[ \delta(\mathbf{x}_\nu(\mathbf{z}_t, t) \,\|\, \tilde{\mathbf{x}}_\theta^{\text{teacher}}(\mathbf{z}_t, t, m/k)) \right], \tag{6}$$

where $\delta$ is a divergence measure (e.g., KLD) and $\tilde{\mathbf{x}}_\theta^{\text{teacher}}(\mathbf{z}_t, t, m/k)$ are the distillation targets. These targets are constructed using $m/k$ sampling steps with the teacher, starting from $\mathbf{z}_t$ and collecting the predicted log-probabilities for each token at the step where a token was denoised. After training, one step of the student should match $m/k$ teacher steps. SDTT can be applied iteratively by reusing the student as teacher in each round (Salimans & Ho, 2022), progressively halving the number of required steps. Empirically, distilling 2 steps per round is most effective.

## 3 Partition Generative Modeling

At each sampling step, MGMs process the entire sequence, including many [MASK] tokens that will not be decoded yet. In contrast, ARMs process clean tokens only, but generate one token at a time. *Partition Generative Models* (PGMs) combine the strengths of both approaches, by generating multiple tokens in parallel, like MGMs, while processing only the clean tokens, like ARMs. *PGMs are a direct extension of the MGM paradigm.* As a result, sampling algorithms, guidance mechanisms, and distillation methods developed for MGMs apply directly to PGMs. Only the neural network architecture must be adapted (Sec. 4).

### 3.1 Training

**From Masking to Partitioning**  Instead of replacing tokens with [MASK] , PGMs partition the sequence into two complementary groups. Given $\mathbf{x} \in \mathcal{D}$ and $t \sim \mathcal{U}[0, 1]$, each token is assigned to group 1 with probability $p_t = 1 - \alpha_t$, and to group 0 otherwise. Let $\mathbf{g} \in \{0, 1\}^L$ denote the group membership vector. We propose a Transformer variant in Sec. 4 that ensures that information cannot flow between groups. Predictions at positions in group 0 depend only on tokens in group 1, and vice-versa (Figure 2). This is consistent with MGMs, where masked tokens are predicted from clean ones, except that PGMs learn from both groups.

**Connection to the MDLM Variational Bound**  In MDLM, the forward process (2) masks each position independently, so at time $t$ an expected fraction $\alpha_t$ of tokens remain clean. PGMs assign an expected fraction $\alpha_t$ of tokens to group 0, which plays the same role as the clean tokens in MDLM. By treating group 0 as clean and group 1 as masked, the MDLM loss weight $w(t) = \frac{\alpha_t'}{1 - \alpha_t}$ is applied to tokens in group 1. By symmetry, tokens in group 0 are weighted by $w(1 - t)$. Therefore, in a single forward pass, PGMs evaluate the MDLM training objective at two complementary masking rates. Hence, the training objective is

$$\mathcal{L}_{\text{PGM}} := \mathbb{E}_{\mathbf{x} \sim \mathcal{D}, t \sim \mathcal{U}[0,1]} \left[ w^{\text{PGM}}(\mathbf{g}, t) \text{CE}(\mathbf{x}_\theta(\mathbf{x}; \mathbf{g}; t), \mathbf{x}) \right], \tag{7}$$

where

$$w^{\text{PGM}}(\mathbf{g}, t)_i = \begin{cases} w(t) & \text{if } \mathbf{g}_i = 0 \\ w(1 - t) & \text{if } \mathbf{g}_i = 1. \end{cases} \tag{8}$$

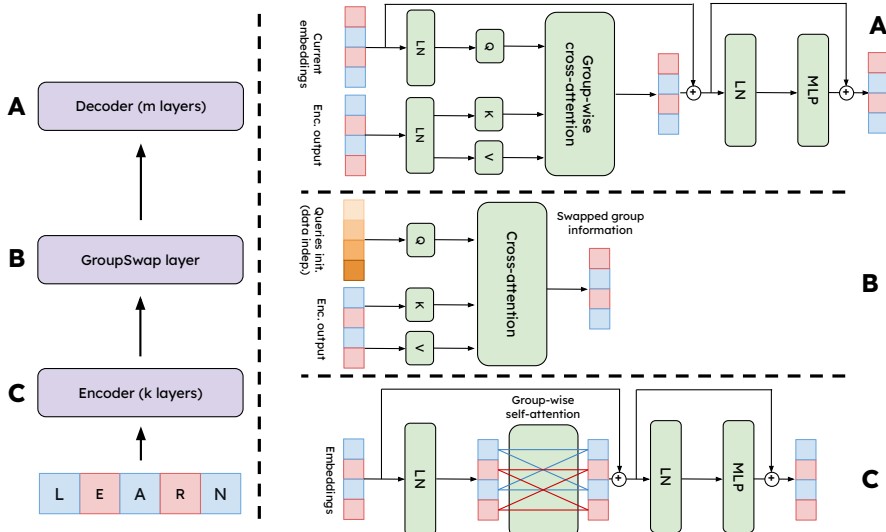

Figure 3: **Partition Transformer.** RoPE is applied before every attention layer (omitted for clarity). **(C)** Encoder: group-wise self-attention (no cross-group flow). **(B)** GroupSwap: cross-attention that routes each position's representation to the opposite group. **(A)** Decoder: group-wise cross-attention to the encoder output (no self-attention).

**Variance Reduction** Unlike MGMs, which compute the loss over masked positions only (Figure 2), PGMs compute the loss at every position, yielding two gradient contributions per training sample. By training on two complementary copies, PGMs reduce the variance. Empirically, training diffusion models with lower variance improves the validation likelihood (Kingma et al., 2023; Sahoo et al., 2024). We study the variance reduction in Sec. 5.3.

## 3.2 SAMPLING

During inference, PGMs process clean tokens only, like ARMs, yet decode tokens in parallel at arbitrary positions, like MGMs (Figure 2). Let $\mathcal{C}_\tau \subseteq \{1, \ldots, L\}$ denote the clean token indices at step $\tau \in \{1, \ldots, T\}$, with $n_\tau = |\mathcal{C}_\tau|$ and $m_\tau = L - n_\tau$. At each step, we select $k_\tau$ masked positions, sample from $p_\theta(\cdot \mid \mathbf{x}_{\mathcal{C}_\tau})$, and add the decoded tokens to $\mathcal{C}_{\tau+1}$. For text, we find that using a fixed schedule with $k_\tau = k$ tokens per step (Algo. 2) improves sample quality and throughput compared to the MDLM posterior that decodes each position with probability $\frac{\alpha_s - \alpha_t}{1 - \alpha_t}$ and requires padding for batched generation (Algo. 3; Suppl. E.2). For images, we experiment with both the confidence and Halton samplers (Suppl. B, Besnier et al. (2025)). The Halton sampler performs better empirically, so we report confidence-based sampler results in Suppl. D.3.

## 4 THE PARTITION TRANSFORMER

PGMs require a careful architectural design. In particular, since our goal is to process a single group only during inference, tokens across groups should not attend to each other. As shown in Figure 2 and Figure 3, we build the *Partition Transformer* such that the predictions for tokens in group 0 are based on tokens in group 1 only. The Partition Transformer implements a mechanism to *swap* the physical location of information across groups. During training, this allows using the input sequence $\mathbf{x}$ as target. During sampling, it moves information about the input tokens from the clean positions to the positions to predict. Our architecture consists of an encoder, a *GroupSwap* layer, and a decoder, which we describe below.

**Encoder** The encoder is made of partition-wise self-attention blocks, which are similar to standard bidirectional transformer blocks except that tokens in separate groups do not attend to each other.

Table 1: Validation perplexity, sampling latency, and throughput (TP) on LM1B and OpenWebText. *PGM $k$ / $m$* uses $k$ encoder and $m$ decoder layers. The best PGM per dataset is highlighted. Latency and TP are measured at batch size 32. † Trained with a $2\times$ larger batch size (Sec. 5.3). See Table 5 for architecture ablations.

| Model | #Params | Val. PPL ↓ | Latency (sec) ↓ | TP (tok/sec) ↑ |
|---|---|---|---|---|
| *LM1B (ctx len. 128)* | | | | |
| MDLM | 170M | 27.67 | 3.78 | 1'081.57 |
| MDLM† (Compl. masking) | 170M | **25.72** | 3.78 | 1'081.57 |
| PGM 6 / 6 | 171M | 26.80 | **2.12** | **1'930.93** |
| *OpenWebText (ctx len. 1024)* | | | | |
| MDLM | 170M | 23.07 | 31.41 | 1'043.22 |
| MDLM† (Compl. masking) | 170M | 22.98 | 31.41 | 1'043.22 |
| PGM 8 / 8 | 203M | 22.61 | **5.86** | **5'585.57** |
| PGM 6 / 6 (dim. 1024) | 268M | **21.43** | 5.93 | 5'518.09 |

**Decoder** The decoder uses cross-attention layers, whose keys and values are computed based on the output of the encoder. In contrast, the queries are computed using either the output of the GroupSwap layer (for the first block of the decoder) or the output of the previous decoder block (see Sec. 4.1). Importantly, *there is no self-attention layer in the decoder*, which allows efficient generation, as we can compute predictions solely at the positions that we will decode.

## 4.1 THE GROUPSWAP LAYER

In the encoder, information remains localized. If a token belongs to group 0, its hidden representation only depends on tokens in group 0. For prediction, however, we require the opposite: representations at positions in group 0 must depend exclusively on group 1, and vice versa. To enforce this, we introduce the *GroupSwap* layer (Figure 3B), which exchanges information between groups. The GroupSwap layer is implemented using cross-attention, and to prevent information leakage, the queries used in cross-attention cannot depend on tokens in the other group. We describe two ways of initializing queries.

**Data-Independent Queries** Let $\mathbf{u} \in \mathbb{R}^H$ be a learnable vector. To initialize the queries, we replicate $\mathbf{u}$ across the sequence length, add fixed positional encodings, and apply layer normalization followed by a linear projection. The query matrix $V \in \mathbb{R}^{L \times H}$ (where $V_{i;\cdot}$ is the $i$-th row) satisfies

$$V_{i;\cdot} = W \left[ \text{LN} \left( u + \text{pos}_{i;\cdot} \right) + b \right], \tag{9}$$

where $W \in \mathbb{R}^{H \times H}$, $b \in \mathbb{R}^H$ are learnable parameters and LN denotes layer normalization (Ba et al., 2016). We use sinusoidal positional encoding (Vaswani et al., 2023):

$$\text{pos}_{i,j} = \begin{cases} \cos\left(\frac{i}{10000^{2j/H}}\right) & \text{if } j < H/2 \\ \sin\left(\frac{i}{10000^{2j/H-1}}\right) & \text{otherwise} \end{cases} \tag{10}$$

**Data-Dependent Queries** Let $X \in \mathbb{R}^{L \times H}$ be the encoder output. We first perform a group-wise aggregation over the sequence length (e.g., `logsumexp` or `mean`) to obtain vectors $Y_0, Y_1 \in \mathbb{R}^H$, the aggregate representations of groups 0 and 1. The queries $V'$ are then

$$V'_{i;\cdot} = V_{i;\cdot} + \begin{cases} Y_1, & \text{if } g_i = 0 \\ Y_0 & \text{otherwise.} \end{cases} \tag{11}$$

## 5 EXPERIMENTS

We compare PGM with MDLM (Sahoo et al., 2024) on standard language modeling datasets, training on LM1B (Chelba et al., 2014) and OpenWebText (OWT; Gokaslan & Cohen (2019)) in Sec. 5.1.

We evaluate them using the validation perplexity and downstream task accuracy before and after distillation with SDTT (Deschenaux & Gulcehre, 2025). We compare PGM with MaskGIT (Chang et al., 2022) on VQGAN-quantized (Esser et al., 2021) ImageNet256 (Deng et al., 2009) (Sec. 5.2). As described in Sec. 3, by predicting each group from the other, PGMs implement a mechanism akin to training on two complementary masked sequences per batch, while also introducing a new architecture (Sec. 4). The effect of complementary masking is studied in isolation in Sec. 5.3. Our experiments show that, for both language and image modeling, and after distillation, PGMs are competitive with MDLM and MaskGIT, while **providing a 5–5.5× throughput improvement for text and a 7.5× improvement for images**. Find more experimental details in Suppl. C.

## 5.1 LANGUAGE MODELING

**Experimental settings**   We closely follow the settings of Sahoo et al. (2024). MDLM uses a modified Diffusion Transformer (Peebles & Xie, 2023; Lou et al., 2024) with RoPE (Su et al., 2023), with 12 layers and an embedding dimension of 768, without time conditioning. We train with a global batch size of 512 for 1M steps, dropout of 0.1, and the Adam optimizer with learning rate $3 \times 10^{-4}$ and no weight decay. We maintain an Exponential Moving Average (EMA) of the weights with decay 0.9999. For PGM, we use the Partition Transformer architecture (Sec. 4) with 12 or 16 layers, embedding dimensions of 768 or 1024, and varying numbers of encoder and decoder layers. On LM1B, all models use a context length of 128, with shorter documents padded and tokenized using the `bert-base-uncased` (Devlin et al., 2019) tokenizer. On OWT, we use a context length of 1024 with sentence packing (Raffel et al., 2023) with the GPT-2 tokenizer and insert an [EOS] token between documents. Since the dataset lacks an official validation split, the last 100k documents are reserved for validation. To evaluate the sample quality, we use the Generative Perplexity (Gen. PPL), computed using GPT-2 Large (Radford et al., 2019), following Sahoo et al. (2024). We cast the logits in `float64` prior to sampling, following Zheng et al. (2025).

**Likelihood Evaluation**   After 1M steps, PGMs with as many layers as MDLM achieve a **validation perplexity of 1.95 lower than MDLM on LM1B** (Table 1). Table 5 (left) shows that balanced models with equal numbers of encoder and decoder layers outperform imbalanced variants. Interestingly, data-independent queries perform comparably to data-dependent queries, so we use the simpler, data-independent version in all subsequent experiments. On OpenWebText, PGMs with the same number of layers and embedding dimension as MDLM slightly underperform (Table 5, right). Increasing the number of encoder and decoder layers by two, or increasing the embedding dimension to 1024, allows PGMs to surpass MDLM in validation perplexity, while **achieving at least 5× higher sampling throughput**. This improved efficiency makes PGMs particularly attractive for scaling test-time computation (Madaan et al., 2023; Yao et al., 2023; Snell et al., 2024; Wu et al., 2024; Chen et al., 2024; Brown et al., 2024; Goyal et al., 2024).

**Downstream Evaluation**   Following Deschenaux & Gulcehre (2024); Nie et al. (2025), we evaluate MDLM and PGMs trained on OpenWebText using the `lm-eval-harness` suite (Gao et al., 2024). As shown in Table 2, PGMs slightly outperform MDLM on six out of eight tasks, although the overall accuracy across models is similar. This suggests that **PGM achieves faster inference without sacrificing downstream performance**. Since `lm-eval-harness` is originally designed for ARMs, we must adapt it for MGMs. Fortunately, both MDLM and PGM can compute a variational bound on the likelihood, which is used in place of the true likelihood to select the most probable answer in multiple-choice tasks. Additional details and tasks are provided in Suppl. D.5.

**Distillation of PGMs**   After likelihood training, PGMs achieve $5 - 5.5\times$ higher throughput than MDLM. To further accelerate sampling, we apply Self-Distillation Through Time (SDTT; Deschenaux & Gulcehre (2025)). To remain as faithful as possible to the implementation of Deschenaux & Gulcehre (2025), we apply the distillation loss to a single group while treating the other as [MASK] tokens. This shows that PGMs are compatible with distillation methods designed for MGMs. We leave the development of new distillation strategies for PGMs to future work. Hence, the setup naturally favors MDLM. Figure 4 (right) and Table 6 compare the Gen. PPL, unigram entropy, and sampling speed of PGM and MDLM. After five rounds of distillation, and with standard ancestral sampling, PGMs achieve higher Generative Perplexity and entropy than MDLM. With nucleus sampling ($p = 0.9$) (Holtzman et al., 2020), PGMs produce samples with comparable perplexity and entropy. Due to

the overhead of nucleus sampling, the speed advantage of PGMs decreases from at least $5\times$ to approximately $4.6\times$ faster than MDLM for the same number of steps (Fig. 4). Generative perplexity alone does not fully capture language model performance, hence we also evaluate distilled models on downstream tasks. As shown in Table 2, distillation slightly shifts accuracy across tasks, but overall performance remains similar. **PGMs still achieve slightly higher accuracy than MDLM on most tasks after distillation**.

## 5.2 Image Modeling

**Experimental Settings** We train MaskGIT (Chang et al., 2022) and PGM on ImageNet256. Images are cropped to a centered square along the longer side and then rescaled to $256 \times 256$. We use the MaskGIT implementation of Besnier et al. (2025), including their pre-trained VQ-GAN tokenizer. We train for 500k steps with a batch size of 256 using AdamW (weight decay 0.03, learning rate 1e-4, cosine schedule with 2500 warmup steps). We use a dropout of 0.1 in the Transformer. All models are class-conditional, with a class-label dropout of 0.1 to enable classifier-free guidance (CFG) at sampling time. As Besnier et al. (2025), we train with one register (Darcet et al., 2024) for the MaskGIT baseline, and two (one per group) for PGM, so that we can use one register during sampling. We sample with the confidence and Halton samplers.

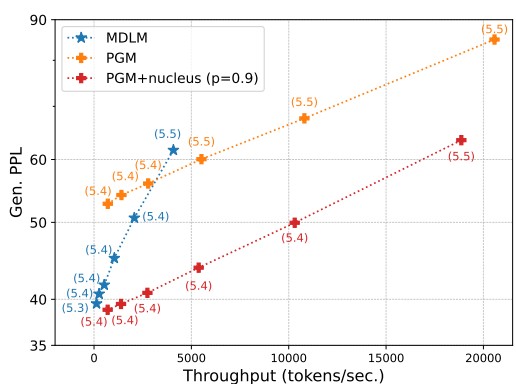

Figure 4: After distillation, PGM (6 / 6, dim. 1024) with nucleus sampling remains significantly faster than MDLM, at matching entropy and Gen. PPL.

**Results** In Figure 1 (left), we compare the Fréchet Inception Distance (FID; Heusel et al. (2018)) of samples from MaskGIT (Chang et al., 2022) and PGM, using the Halton sampler and classifier-free guidance with the guidance weight $w \in \{0, 1, \dots, 6\}$ that yields the lowest FID. PGM 12/12 achieves a $7.5\times$ higher throughput with only a slight FID degradation (5.54 vs. 5.35). **Increasing the sampling steps to 64 further improves the FID to 4.56, while remaining $3.9\times$ faster than MaskGIT**. See Suppl. D.3 for full results across guidance strengths.

## 5.3 Isolating the Effect of Complementary Masking

**Experimental Setup** To disentangle the contributions of PGM, we isolate the effect of complementary masking (Sec. 3) by training a standard bidirectional Transformer with double the batch size. Each input sequence is turned into two complementary masked copies: if the token at position $\ell$ is masked in one copy, it remains unmasked in the other. This setup provides an upper bound on the potential gains, as it directly measures the benefit of complementary masks during training.

**Results** Table 1 shows that complementary masking improves the validation perplexity on LM1B and OWT, though with smaller gains on OWT. On both datasets, a gap remains between PGM and MDLM with complementary masking. This suggests that the current neural network architecture can be improved further. Because of the smaller improvement on OWT, we must increase the parameter count to surpass MDLM. Nonetheless, recall that despite having more parameters, PGMs remain at least $5\times$ faster than MDLM during sampling. In Suppl. D.1, we present preliminary experiments exploring why complementary masking improves performance on LM1B but not on OpenWebText.

## 6 Related Work

**Discrete Diffusion** Although autoregressive models currently dominate text generation, recent advances in discrete diffusion (Austin et al., 2023; Lou et al., 2024; Shi et al., 2025; Sahoo et al., 2024; von Rütte et al., 2025; Schiff et al., 2025; Haxholli et al., 2025; Sahoo et al., 2025a) and discrete flow matching (Campbell et al., 2024; Gat et al., 2024) have demonstrated that MGMs can

Table 2: **Accuracy on downstream tasks** (Gao et al., 2024). HS: HellaSwag, OQA: OpenBook QA. Arc: Arc-easy. We select the tasks following Nie et al. (2025). We see that distillation slightly changes the downstream tasks performance, but that PGMs continue to outperform MDLM on most tasks. The best performance is **bolded**, while the second best is underlined.

|  | LAMBADA | Arc | BoolQ | HS | OQA | PIQA | RACE | SIQA |
|---|---|---|---|---|---|---|---|---|
| *Before Distillation* | | | | | | | | |
| MDLM | 38.52 | 37.88 | 49.42 | 31.36 | **28.60** | 58.27 | **28.04** | 38.84 |
| PGM 8 / 8 | **46.98** | **40.40** | **53.49** | 33.20 | 26.60 | 58.92 | 26.89 | 39.97 |
| PGM 6 / 6 (1024) | 41.39 | 39.98 | 49.82 | **34.27** | 25.40 | **59.19** | 27.37 | **40.28** |
| *After Distillation (SDTT)* | | | | | | | | |
| MDLM | 41.34 | 33.80 | 48.59 | 30.75 | **28.80** | 57.73 | 27.94 | 38.79 |
| PGM 8 / 8 | **47.22** | **37.42** | **51.50** | 31.62 | 25.80 | 59.03 | **30.62** | **39.61** |
| PGM 6 / 6 (1024) | 44.48 | 36.70 | 49.36 | **32.55** | 25.00 | **59.85** | 27.37 | 39.25 |

approach AR models in generation quality. We propose a simple framework that allows sampling without processing any [MASK] tokens, but remains compatible with methods developed for MGMs (such as distillation and alternative samplers).

**Variable Length Masked Diffusion** Block Diffusion (BD; Arriola et al. (2025)) enables partial KV-caching (Pope et al., 2022) by generating tokens block-by-block using discrete diffusion. BD improves throughput but sacrifices the any-order generation capabilities of MGMs. We do not experiment with integrating causal attention to enable KV-caching. However, Ma et al. (2025); Wu et al. (2025) show that KV caching can be integrated post-hoc into MGMs despite being trained without causal attention. FlexMDM (Kim et al., 2025) and Edit Flows (Havasi et al., 2025) enable variable-length generation via insertion, deletion, and replacement. While promising, these depart from the simplicity of MGM and PGM. Finally, Eso-LMs (Sahoo et al., 2025b) train with a hybrid AR-MGM objective. Sahoo et al. (2025b) first sample a draft in MGM mode, then fill in the remaining tokens autoregressively. During training, Eso-LMs must choose the fraction of examples to process in AR versus MGM mode, which adds a hyperparameter to tune. Eso-LMs use [MASK] tokens during training in MGM mode, whereas PGMs do not because of the Partition Transformer.

**Non-Autoregressive Language Models** Any-order and any-subset autoregressive models (Yang et al., 2020; Pannatier et al., 2024; Shih et al., 2022; Guo & Ermon, 2025) factorize the sequence distribution autoregressively over permutations of tokens. Hence, these models use causal attention and generate tokens one by one. In contrast, MGMs use bidirectional attention and generate multiple tokens in parallel, which is the setting PGM builds on.

## 7 CONCLUSION

We introduce Partition Generative Modeling (PGM), a novel approach to masked generative modeling that eliminates [MASK] tokens entirely. PGM achieves significant improvements in inference speed on both text and images, with minimal effect on quality. The significant improvements suggest that PGM might be suited for domains that benefit from test-time scaling, such as coding and reasoning. We show that PGMs can be distilled for further acceleration. Future work should explore optimizations to the PGM architecture, investigate distillation techniques specifically designed for PGMs, and extend the approach to multimodal settings. In summary, PGM offers an alternative to masked generative models, with particular advantages for applications where inference speed is critical.

## 8 ACKNOWLEDGEMENTS

This work has received funding from the Swiss State Secretariat for Education, Research and Innovation (SERI). We are grateful to Razvan Pascanu, Sungjin Ahn, Jaesik Yoon, Mingyu Jo, Subham Sahoo and Zhihan Yang for insightful discussions and suggestions. We acknowledge the SCITAS team at EPFL for providing access to their cluster, and the Swiss National Supercomputing Centre for the Alps platform. We are grateful to Karin Gétaz for her administrative assistance.

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

---

**Algorithm 1** Building the Halton Unmasking Schedule

---

1: **Input:** Grid size $H$
2: **Output:** Ordered list of $L = H^2$ grid positions
3: schedule $\leftarrow$ []
4: seen $\leftarrow \emptyset$
5: $i \leftarrow 1$
6: **while** $|\text{schedule}| < L$ **do**
7:     cell $\leftarrow (\lfloor \Phi_2(i) \cdot H \rfloor, \lfloor \Phi_3(i) \cdot H \rfloor)$
8:     **if** cell $\notin$ seen **then**
9:         seen $\leftarrow$ seen $\cup$ {cell}
10:         schedule.append(cell)
11:     **end if**
12:     $i \leftarrow i + 1$
13: **end while**
14: **return** schedule

---

## A    LIMITATIONS

To match the validation perplexity of the MDLM baseline at a context length of 1024, our models require a slight increase in parameters. We attribute this to the GroupSwap layer, and future work will explore more efficient mechanisms for information exchange between groups in PGMs. While PGMs offer faster inference, their training is slightly more computationally expensive (Appendix E), as we use `torch`'s default attention implementation ("sdpa") for simplicity. By reordering tokens according to their group assignment, the self-attention matrices becomes block-diagonal. Future work will explore efficient kernel implementations that exploit this block-diagonal sparsity. Partition Generative Modeling is a general framework, and its application to multimodal settings remains an open direction for future research.

## B    SAMPLING WITH HALTON SEQUENCES (MASKGIT)

Let $\mathcal{S} = \{\ell \mid z_\tau^\ell = [\text{MASK}]\}$ denote the set of masked positions at step $\tau$. MGMs samples independently at all the masked positions from a product of marginal predictions $\prod_{\ell \in \mathcal{S}} p_\theta(\mathbf{x}^\ell \mid \mathbf{z}_\tau)$, not from the joint $p(\mathbf{x} \mid \mathbf{z}_\tau)$. The KL divergence between the joint and the product of marginals is the mutual information (MI) (Besnier et al., 2025):

$$D_{\text{KL}}\left(p(\{x^\ell\}_{\ell \in \mathcal{S}} \mid \mathbf{z}_\tau) \,\Big\|\, \prod_{\ell \in \mathcal{S}} p_\theta(x^\ell \mid \mathbf{z}_\tau)\right) = \text{MI}(\{x^\ell\}_{\ell \in \mathcal{S}} \mid \mathbf{z}_\tau). \tag{12}$$

Empirically, the denoiser is most confident at positions close to previously generated tokens. Therefore, the decoded tokens tend to cluster together. This leads to a larger MI, with a higher risk of generating inconsistent tokens. Besnier et al. (2025) propose to replace the confidence-based ordering with a *Halton sequence* (Halton, 1964), a low-discrepancy sequence whose consecutive points are far apart in space. Formally, let $a_j(i)$ denote the $j$-th digit of the representation of $i \in \mathbb{N}_+$ in base $b$, so that $i = \sum_j a_j(i) b^j$. Let the *radical-inverse function* $\Phi_b$ denote the inverse of $i$ in base $b$:

$$\Phi_b(i) = \sum_{j=0}^m a_j(i) b^{-(j+1)} \in [0, 1). \tag{13}$$

Recall that MaskGIT operates over a square grid of VQGAN tokens (Esser et al., 2021), of size $H \times H$ ($L = H^2$). To build the unmasking order (Algo. 1), Besnier et al. (2025) iterate over positive integers $i$ and compute the cell coordinates $(\lfloor \Phi_2(i) \cdot H \rfloor, \lfloor \Phi_3(i) \cdot H \rfloor)$. If the cell was already visited for some $i' < i$, it is skipped. This continues until all $L$ cells are visited. The Halton scheduler consistently improves the FID and, unlike confidence-based sampling, continues to improve with more inference steps (Besnier et al., 2025). In our image experiments, we use both confidence and Halton schedulers.

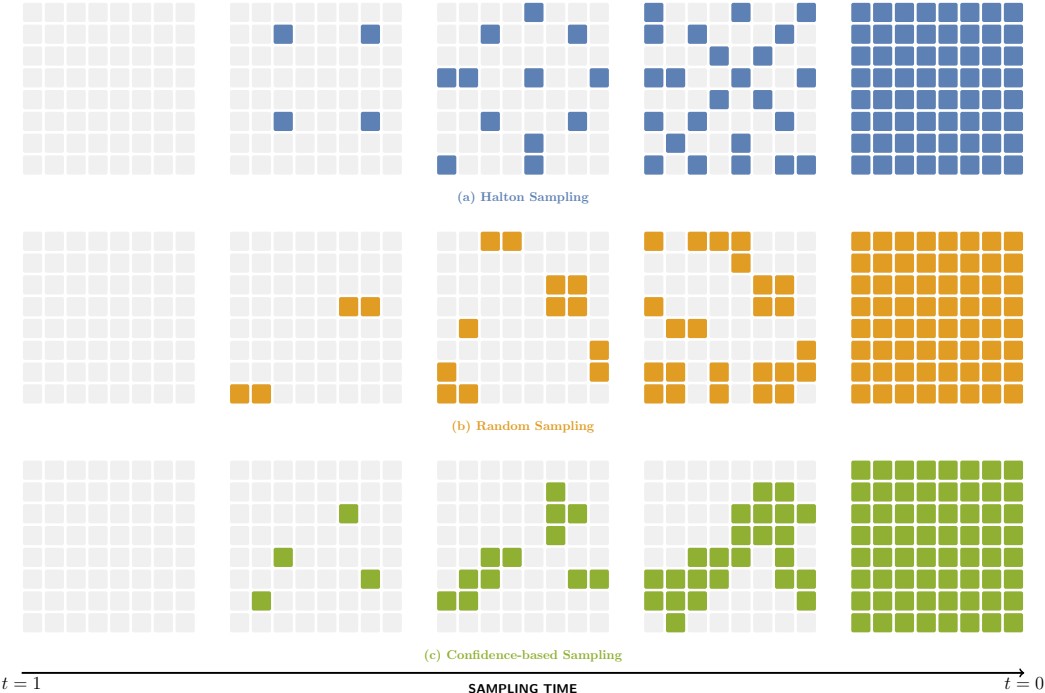

Figure 5: Comparison of unmasking schedules on a 2D grid. **(a)** Halton sampling covers space uniformly. **(b)** Random sampling can leave large areas empty. **(c)** With confidence-based sampling, new tokens are decoded close to existing ones, leading to high mutual information and a risk of inconsistent generation.

## C    EXPERIMENTAL DETAILS

We trained all models from scratch. Our baselines achieve similar performance as reported by Sahoo et al. (2024). On LM1B, we obtain a validation perplexity of 27.67 after 1M steps (compared to MDLM's reported 27.04), while on OWT, we reach 23.07 (versus MDLM's 23.21).

Minor differences can be expected since estimating the perplexity of diffusion language models involves a Monte-Carlo approximation of the NELBO (1) with finitely many samples. Although we used libraries (e.g., PyTorch) with the same version as MDLM, differences in compute environments and underlying software stacks may also contribute to these variations. Since the performance gap is small, we are confident that we used the code of MDLM correctly.

### C.1    LM1B

For the LM1B dataset, we employed the `bert-base-uncased` tokenizer with a context length of 128 tokens, padding shorter sequences. Our architecture consisted of a Diffusion Transformer (DiT) with 12 transformer blocks, 12 attention heads, a hidden dimension of 768, and a dropout rate of 0.1. We optimized the model using Adam (Kingma & Ba, 2017) (learning rate 3e-4, betas of (0.9, 0.999), epsilon 1e-8) without weight decay. We based our implementation on the official MDLM codebase. We trained with a global batch size of 512 across 8 GPUs (2 nodes with 4 GPUs), gradient clipping at 1.0, and a constant learning rate with 2,500 steps of linear warmup. We trained for 1 million steps with an EMA rate of 0.9999. Besides the neural network hyperparameters, the other parameters were unchanged when training the PGM.

### C.2    OWT

For the OpenWebText (OWT) dataset, we used the GPT-2 tokenizer with a context length of 1024 tokens. Our architecture consisted of a Diffusion Transformer (DiT) with 12 transformer blocks,

Table 3: Latency and throughput for a single forward+backward pass of the MDLMs and PGMs, computed on a single A100-SXM4-80GB GPU. On LM1B, PGM introduces a negligible overhead over MDLM. On OWT, our PGM with 6 encoder and decoder layers and an embedding dimension of 1024 achieves around 75% of the training throughput of MDLM. Recall that at inference, the same PGM is around $5\times$ faster than MDLM.

| Model | Forward Pass | | Forward + Backward | |
|---|---|---|---|---|
| | Latency (ms) | Seq/sec | Latency (ms) | Seq/Sec |
| *LM1B (context length 128, batch size 64, trained on 8 GPUs)* | | | | |
| MDLM | $0.03 \pm 0.00$ | $1'978.87 \pm 44.21$ | $0.08 \pm 0.00$ | $714.80 \pm 15.47$ |
| PGM 6 / 6 | $0.03 \pm 0.00$ | $1'966.60 \pm 102.14$ | $0.08 \pm 0.00$ | $794.42 \pm 18.81$ |
| *OpenWebText (context length 1024, batch size 32, trained on 16 GPUs)* | | | | |
| MDLM | $0.13 \pm 0.00$ | $233.28 \pm 2.58$ | $0.39 \pm 0.00$ | $80.86 \pm 0.15$ |
| PGM 8 / 8 | $0.17 \pm 0.00$ | $188.07 \pm 0.75$ | $0.47 \pm 0.00$ | $68.04 \pm 0.08$ |
| PGM 6 / 6 (dim. 1024) | $0.18 \pm 0.00$ | $176.47 \pm 0.65$ | $0.50 \pm 0.00$ | $62.85 \pm 0.19$ |

12 attention heads, a hidden dimension of 768, and a dropout rate of 0.1. We optimized the model using Adam (Kingma & Ba, 2017) with a learning rate of 3e-4, betas of (0.9, 0.999), and epsilon of 1e-8, without weight decay. We trained with a global batch size of 512 across 16 GPUs (4 nodes with 4 GPUs). We applied gradient clipping at 1.0 and used a constant learning rate schedule with 2,500 steps of linear warmup. The model was trained for 1 million steps with an EMA rate of 0.9999. Besides the neural network hyperparameters, the other parameters were unchanged when training the PGM.

## C.3 IMAGENET

For the ImageNet experiments, we used a pre-trained VQGAN tokenizer (Esser et al., 2021; Besnier et al., 2025), following the setup of Besnier et al. (2025). The images are tokenized into sequences of 1024 tokens. This allowed for a direct comparison between PGM and MaskGIT, both trained in the codebase of Besnier et al. (2025) and the FID is evaluated using the Halton sampler and the confidence sampler. We compute the FID between 50k generated images and the validation set, following Besnier et al. (2025)

All models use 24 transformer blocks. For PGM, we add a GroupSwap layer to enable information exchange between partition groups. We use the same hyperparameters as HaltonMaskGIT for all models, except we reduce the training duration to 500k steps (from 2M) due to computational constraints. All models are trained to be class-conditional, which enables the use of classifier-free guidance to significantly improve performance.

## C.4 IMPACT OF NUMERICAL PRECISION ON SAMPLING

Zheng et al. (2025) identified that Masked Diffusion Models often achieve lower Generative Perplexity results because of underflow in the logits when sampling using low precision. The resulting decrease in token diversity can make evaluations based solely on Generative Perplexity misleading. Hence, we always cast the logits to FP64 before sampling.

## C.5 SAMPLE-BASED EVALUATION

**Generative Perplexity**    We use the Generative Perplexity to evaluate the quality of samples, following prior work (Lou et al., 2024; Sahoo et al., 2024; Deschenaux & Gulcehre, 2025). The Generative Perplexity measures how well a reference model (in our case, GPT-2 Large) can predict the next token in generated sequences. Specifically, we generate $1'024$ samples from each model being evaluated. For each generated sample, we compute the Generative Perplexity using GPT-2 Large as follows:

$$\text{Perplexity} = \exp\left(-\frac{1}{L}\sum_{i=1}^{L}\log p_{\text{GPT-2 Large}}(x_i|x_{<i})\right), \tag{14}$$

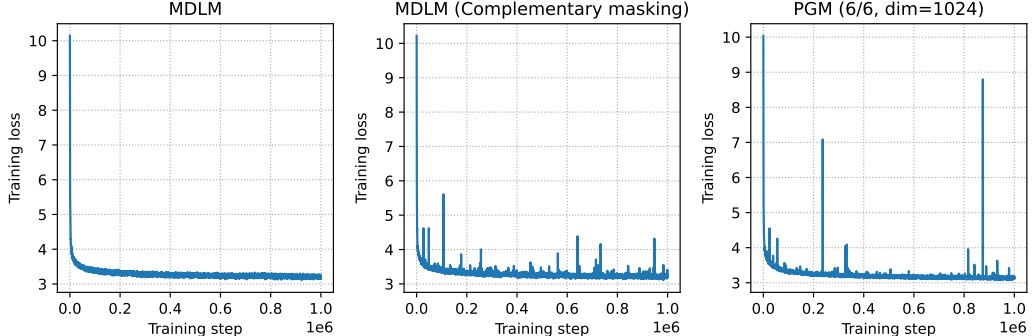

Figure 6: Training loss of MDLM, MDLM with Complementary Masking (Section 5.3) and PGM. Complementary masking seems to introduce spikes in the loss, even though it did not cause the models to diverge.

where $L$ is the length of the sequence, $x_i$ is the $i$-th token, and $p_{\text{GPT-2 Large}}(x_i|x_{<i})$ is the probability assigned by GPT-2 Large to token $x_i$ given the preceding tokens $x_{<i}$.

**Unigram Entropy**   Unfortunately, a low Generative Perplexity can be achieved by generating repetitive text. To catch such cases, we compute the average unigram entropy of the generated samples:

$$\text{Unigram Entropy} = -\frac{1}{N}\sum_{i=1}^{N}\sum_{v\in\mathcal{V}}\frac{c(v,\mathbf{x}^{(i)})}{L}\log\frac{c(v,\mathbf{x}^{(i)})}{L}, \tag{15}$$

where $\mathcal{V}$ is the vocabulary, $v$ is a token of the vocabulary, and $c(v,\mathbf{x})$ is the empirical appearance count of the token $v$ in the sequence $\mathbf{x}$. Low unigram entropy helps us to catch degenerate generation, as shown by Dieleman et al. (2022).

**Fréchet Inception Distance and Inception Score**   On image generation tasks, we evaluate the quality of samples using the Fréchet Inception Distance (FID) (Heusel et al., 2018) and Inception Score (IS) (Salimans et al., 2016). Both metrics are computed using $50'000$ images, following the standard practice.

## D   ADDITIONAL RESULTS

### D.1   IMPACT OF CONTEXT LENGTH ON THE EFFECTIVENESS OF COMPLEMENTARY MASKING

There are three key differences between our experiments on LM1B and OWT. First, we used different tokenizers: `bert-base-uncased` for LM1B and `GPT2`'s tokenizer for OWT, following the setup of MDLM (Sahoo et al., 2024). Second, the context lengths differ significantly: 128 tokens for LM1B versus 1024 for OWT. Third, we train on different datasets that might have different characteristics.

We observed that complementary masking helps when training on OWT using a shorter context length of 128 tokens with the GPT-2 tokenizer. Indeed, after 200k training step, the MDLM with complementary masking achieved a validation PPL of 37.92, outperforming the standard MDLM, which reached 39.90. This suggests that PGMs may not need extra parameters when the sequence length is short. Exploring the use of PGMs in domains where the sequence length is short, such as modeling chemical sequences, is a promising direction for future work.

### D.2   MDLM+SDTT VS PGM+SDTT

The precision of logits during sampling can have a significant effect on sample quality, as noted in Appendix C.4. Hence, we cast all logits to FP64 prior to sampling, unlike the original MDLM and SDTT implementations.

Using higher precision also affects distillation, which compresses two sampling steps into one. As shown in Table 7, models distilled with float32 achieve lower Generative Perplexity than those trained with mixed precision (bfloat16). We therefore report float32 results in the main body.

### D.3 ADDITIONAL RESULTS ON IMAGENET

Table 8 and Table 9 show the FID, IS, latency, and throughput for the Confidence and Halton samplers. Overall, the Halton sampler works best for both MaskGIT and PGM. With 32 steps and the confidence-based sampler, PGM 12/12 gets a better FID than MaskGIT and is $3.58\times$ faster. With 32 steps and the Halton sampler, PGM has a slightly higher FID than MaskGIT (5.54 vs 5.35), but is $7.5\times$ faster. If we increase the number of sampling steps to 64, PGM achieves an FID of 4.56, which is better than MaskGIT, and is $3.92\times$ faster. Generally, the 12/12 variant outperforms the 14/10 variant, which suggests that balanced number of layers in the encoder and decoder is beneficial, just as for language modeling (Table 5).

### D.4 TRAINING STABILITY

Complementary masking introduces occasional spikes in the training loss in both MDLMs and PGMs, as shown in Figure 6. This phenomenon should be kept in mind when scaling PGMs to larger sizes. Despite these spikes, all runs converged on the first attempt. We observed different precision requirements between models. For loss computations, MDLMs performed best with BF16 precision, while PGMs achieved better results with FP32 precision. Both models use mixed precision within the neural network; the precision difference only affects computations performed outside the model, such as the loss calculation.

### D.5 ADDITIONAL DOWNSTREAM TASKS

Table 4 reports additional downstream results as in Deschenaux & Gulcehre (2025), where PGM outperforms MDLM on all but one benchmark, with only a small gap on the latter. We evaluate models with the `lm-eval-harness` library (Gao et al., 2024), originally designed for autoregressive LMs and adapted here for MDLM. For multiple-choice tasks, `lm-eval-harness` computes the log-likelihood of each candidate answer $\mathbf{y}_i$ given a prefix $\mathbf{x}$, i.e., $p(\mathbf{y}_i|\mathbf{x})$, and selects the answer with the highest score.

While `lm-eval-harness` uses the log-likelihood of the continuation, the NELBO objective (1) bounds the log-likelihood of the *complete* sequence $(\mathbf{x}, \mathbf{y}_i)$. However, we only need to know which continuation achieves the highest log-likelihood, not to compute the exact log-likelihood. Using Bayes' theorem, we note that

$$\log p(\mathbf{y}_i|\mathbf{x}) = \log p(\mathbf{x}, \mathbf{y}_i) - \log p(\mathbf{x}) \propto \log p(\mathbf{x}, \mathbf{y}_i), \tag{16}$$

since $\log p(\mathbf{x})$ is constant with respect to $\mathbf{y}_i$. Therefore, we can simply evaluate the variational bound on $\log p(\mathbf{x}, \mathbf{y}_i)$ to select the most likely continuation $y_i$.

### D.6 PERFORMANCE ON LONGER CONTEXT LENGTH

Due to the high computational cost, we were unable to train models with context lengths greater than 1024. Nevertheless, we report the latency and throughput of both MDLM and PGM at a context length of 4096. As shown in Table 10, PGM remains substantially faster than MDLM in this setting.

## E COMPUTATIONAL COSTS

This section presents the computational costs associated with the models reported in this paper. We exclude costs associated with exploratory experiments that yielded inferior results and were not included in this manuscript.

### E.1 TRAINING COSTS

Training PGMs is currently slower than training MGMs since we use `torch.sdpa` with dense tensor masks. Future work should explore efficient kernels to address this limitation. We measure the

| Model | LAMBADA | ARC-e | ARC-c | HSwag | MathQA | PIQA | WinoG |
|---|---|---|---|---|---|---|---|
| MDLM | 38.52 | 34.26 | **24.66** | 31.54 | 20.70 | 57.89 | 51.93 |
| PGM 8 / 8 | **46.98** | 37.37 | 24.06 | 33.10 | 21.24 | 59.09 | 51.30 |
| PGM 6 / 6 (1024) | 41.39 | **38.80** | 22.95 | **33.92** | **21.71** | **61.43** | **54.30** |

Table 4: Accuracy on downstream tasks. We evaluate MDLM and PGM on LAMBADA, ARC Easy and Challenge, HellaSwag, MathQA, PIQA, and WinoGrande. Both models show comparable performance across tasks. PGM outperforms MDLM on all but one benchmark, where the difference between MDLM and PGM 8 / 8 is small.

latency and throughput using a single NVIDIA A100-SXM4-80GB GPU, with results reported in Table 3. We compute the mean and standard deviation over 100 batches after 2 warmup batches.

The total training duration approximately equals the per-step latency multiplied by the number of steps. Experiments with complementary masking required twice the computational resources due to larger batch sizes and gradient accumulation. Training times for 1M steps varied by dataset: approximately 22 hours for LM1B, 4.5 days for OWT, and 3.8 days for ImageNet.

Despite the current training overhead, we are confident that future work can improve the training efficiency of PGMs, thanks to their block-diagonal attention patterns, once the tokens are grouped together along the sequence-length axis.

### E.2 INFERENCE COSTS

We evaluate the inference efficiency of PGMs compared to MDLMs and GPT-2 with KV caching. As shown in Figure 1, PGMs achieve around $5 - 5.5\times$ improvements in throughput over MDLM while reaching superior Generative Perplexity. For inference measurements, we use a single NVIDIA A100-SXM4-80GB GPU. The efficiency gain stems from the ability of PGMs to process only unmasked tokens during inference, as illustrated in Figure 2. Table 6 compares MDLM and PGMs on the Generative Perplexity, unigram entropy, latency, and throughput. We compute the mean and standard deviation of the latency and throughput over 20 batches after two warmup batches.

### E.3 LICENSING

Our code and model artifacts will be released under the MIT license. The OWT dataset (Gokaslan & Cohen, 2019) is available under the Apache License 2.0. We were unable to identify a specific license for the LM1B dataset (Chelba et al., 2014). The images in ImageNet remain the property of their respective copyright holders.

---

**Algorithm 2** Simplified Sampling for PGMs

---

1: **Input:** Batch size BS, number of steps K, model length L, special BOS index
2: **Output:** Generated samples x
3: x ← empty_tensor(BS, 1)                                    ▷ *Initialize*
4: x[:, 0] ← BOS                                    ▷ *Set BOS as first token*
5: k ← L/K                                    ▷ *Number of tokens to denoise at each step*
6: decoded_positions ← zeros(BS, 1)     ▷ *Keep track of already-decoded and positions to decode*
7: positions_to_decode ← 1+ rand_row_perm(BS, L-1) ▷ *Each rows is a permutation of* $\{1, ..., L\}$
8: **for** _ in range(K) **do**
9:     pos_to_decode ← positions_to_decode[:, :k]            ▷ *Random positions to be predicted*
10:     new_values ← pgm_predict(x, decoded_positions, pos_to_decode)
11:     $x$ ← concat([x, new_values], dim=1)     ▷ *Add new values to the sequence length dimension*
12:     decoded_positions ← concat([decoded_positions, pos_to_decode], dim=1)
13:     positions_to_decode ← positions_to_decode[:, k:]        ▷ *Remove the k decoded positions*
14: **end for**
15: out ← reorder(x, decoded_positions)                    ▷ *Sort based on positions*
16: **return** out

---

---

**Algorithm 3** MDLM-equivalent sampling for PGMs.

---

1: **Input:** Batch size BS, number of steps K, model length L, special BOS index
2: **Output:** Generated samples x
3: x ← empty_tensor(BS, 1)                                                            ▷ *Initialize*
4: x[:, 0] ← BOS                                                          ▷ *Set BOS as first token*
5: k ← L/K                                              ▷ *Number of tokens to denoise at each step*
6: clean_positions ← zeros(BS, 1)                        ▷ *Keep track of clean and noisy positions*
7: concrete_lengths ← ones(BS, 1) ▷ *Keep track of the actual length of each sequence (some are padded).*
8: noisy_positions ← 1+ rand_row_perm(BS, L-1)
9: **for** _ in range(K) **do**
10:      n_denoise_per_seq, noisy_pos_input ← **sample_noisy**(noisy_positions, k)      ▷ *Algorithm Algo. 4*
11:      new_values ← pgm_predict(x, clean_positions, noisy_pos_input)
12:      x, clean_positions, noisy_positions, concrete_lengths ← **extract_predictions**(
13:          x,                                                              ▷ *Algorithm Algo. 5*
14:          clean_positions,
15:          noisy_positions,
16:          noisy_pos_input,
17:          concrete_lengths,
18:          n_denoise_per_seq,
19:          new_values)
20: **end for**
21: out ← reorder(x, clean_positions)                               ▷ *Sort based on clean_positions*
22: **return** out

---

**Algorithm 4** Sample the number of tokens to denoise from a binomial distribution and pad the input.

---

1: **Input:** Noisy positions tensor, probability of denoising prob_denoise, model length L, concrete lengths tensor
2: **Output:** Noisy positions to denoise
3: n_denoise_per_seq ← binomial(BS, L, prob_denoise)      ▷ *Sample from binomial distribution*
4: n_denoise_per_seq ← min(n_denoise_per_seq, L - concrete_lengths)      ▷ *Don't denoise more than available*
5: denoise_seq_len ← max(n_denoise_per_seq, 0)          ▷ *Maximum number of tokens to denoise*
6: **if** denoise_seq_len = 0 **then**
7:      **return** empty_tensor()                                           ▷ *Nothing to denoise*
8: **end if**
9: noisy_pos_input ← noisy_positions[:, :denoise_seq_len]      ▷ *Some predictions won't be used*
10: **return** n_denoise_per_seq, noisy_pos_input

---

---

**Algorithm 5** Extract the correct number of predictions per sequence

---

1: **Input:** x, concrete_lengths, n_denoise_per_seq, denoised_token_values, clean_positions, noisy_positions, noisy_pos_input
2: **Output:** Updated x, clean_positions, noisy_positions, concrete_lengths
3: new_concrete_lengths ← concrete_lengths + n_denoise_per_seq      ▷ *Update sequence lengths*
4: n_tok_to_add ← max(new_concrete_lengths) - shape(x, 1)      ▷ *Calculate padding needed*
5: **if** n_tok_to_add > 0 **then**
6:     pad ← zeros(BS, n_tok_to_add)      ▷ *Create padding tensor*
7:     x ← concat(x, pad, dim=1)      ▷ *Pad the sequences*
8:     clean_positions ← concat(clean_positions, pad, dim=1)      ▷ *Pad the positions*
9: **end if**
10: **for** i in range(BS) **do**
11:     **if** n_denoise_per_seq[i] = 0 **then**
12:         continue      ▷ Skip if no tokens to denoise
13:     **end if**
14:     x[i, concrete_lengths[i]:new_concrete_lengths[i]] ←
15:         denoised_token_values[i, :n_denoise_per_seq[i]]
16:     clean_positions[i, concrete_lengths[i]:new_concrete_lengths[i]] ←
17:         noisy_pos_input[i, :n_denoise_per_seq[i]]
18:     noisy_positions[i, :shape(noisy_positions, 1) - n_denoise_per_seq[i]] ←
19:         noisy_positions[i, n_denoise_per_seq[i]:]
20: **end for**
21: **return** x, clean_positions, noisy_positions, new_concrete_lengths

---

| Model (LM1B) | Val. PPL ↓ |
|---|---|
| *200k steps* | |
| MDLM | 34.29 |
| MDLM (Compl. masking) | **30.87** |
| PGM 8 / 4 | 32.83 |
| PGM 10 / 2 | 33.55 |
| PGM 4 / 8 | 32.84 |
| PGM 6 / 6 | 32.69 |
| PGM 6 / 6 (lsm) | 32.70 |
| PGM 6 / 6 (mean) | 33.89 |
| *1M steps* | |
| MDLM | 27.67 |
| MDLM (Compl. masking) | **25.72** |
| PGM 6 / 6 | 26.80 |

| Model (OWT) | Val. PPL ↓ |
|---|---|
| *200k steps* | |
| MDLM | 25.35 |
| MDLM (Compl. masking) | 25.32 |
| PGM 6 / 6 | 26.96 |
| PGM 8 / 8 | 25.10 |
| PGM 10 / 6 | 25.19 |
| PGM 6 / 6 (dim. 1024) | **23.75** |
| *1M steps* | |
| MDLM | 23.07 |
| MDLM (Compl. masking) | 22.98 |
| PGM 8 / 8 | 22.61 |
| PGM 6 / 6 (dim. 1024) | **21.43** |

Table 5: Perplexity evaluations. Validation perplexity of the Masked Diffusion Language Model (MDLM) and PGMs (ours) on LM1B and OpenWebText (OWT). The row *MDLM (Compl. masking)* denotes an MDLM trained with the complementary masking strategy discussed in Section 5.3. The row *PGM k / m* denotes a PGM with $k$ encoder and $m$ decoder layers, and we highlighted the best PGM results in gray. *lsm* and *mean* denote the *logsumexp* and *mean* queries initializations (Section 4). **Takeaway:** using the same number of layers in the encoder and decoder, and data-independent queries performed best. On LM1B, our PGM reaches 1.95 lower perplexity than MDLM after 1M steps. On OWT, we grow the embedding dimension or the number of layers to outperform MDLM on OWT.

Table 6: Sample quality and efficiency on OpenWebText with different numbers of sampling steps. We generate sequences of 1024 tokens with a batch size of 32 to measure the latency and throughput. PGM 6 / 6 with a hidden dimension of 1024 and uniform sampling achieves at least a $5\times$ latency and throughput improvement over MDLM, with better Generative Perplexity and matching entropy.

| Model | Gen. PPL ↓ | Entropy ↑ | Latency ↓ (ms) | Throughput ↑ (tok/s) |
|---|---|---|---|---|
| *MDLM* | | | | |
| 32 steps | 192.31 | 5.73 | $8.037 \pm 0.01$ | $4'077.08 \pm 3.06$ |
| 64 steps | 142.58 | 5.69 | $15.82 \pm 0.01$ | $2'070.67 \pm 0.69$ |
| 128 steps | 122.89 | 5.67 | $31.41 \pm 0.01$ | $1'043.22 \pm 0.16$ |
| 256 steps | 113.96 | 5.66 | $62.54 \pm 0.01$ | $523.90 \pm 0.06$ |
| 512 steps | 109.05 | 5.64 | $124.94 \pm 0.16$ | $262.26 \pm 0.33$ |
| 1024 steps | 106.75 | 5.64 | $249.31 \pm 0.11$ | $131.42 \pm 0.05$ |
| *PGM 8 / 8 (uniform sampling)* | | | | |
| 32 steps | 189.02 | 5.73 | $1.55 \pm 0.01$ | $21'120.99 \pm 83.59$ |
| 64 steps | 143.79 | 5.69 | $3.00 \pm 0.01$ | $10'914.91 \pm 41.69$ |
| 128 steps | 122.21 | 5.66 | $5.86 \pm 0.02$ | $5'585.57 \pm 24.49$ |
| 256 steps | 112.48 | 5.65 | $11.64 \pm 0.03$ | $2'814.99 \pm 9.33$ |
| 512 steps | 108.76 | 5.64 | $22.98 \pm 0.02$ | $1'425.89 \pm 1.61$ |
| 1024 steps | 107.03 | 5.63 | $45.84 \pm 0.03$ | $714.71 \pm 0.50$ |
| *PGM 8 / 8 (non uniform sampling)* | | | | |
| 32 steps | 194.09 | 5.73 | $2.07 \pm 0.02$ | $15'764.09 \pm 192.12$ |
| 64 steps | 143.60 | 5.69 | $3.90 \pm 0.07$ | $8'405.14 \pm 158.01$ |
| 128 steps | 124.38 | 5.67 | $7.41 \pm 0.08$ | $4'419.77 \pm 53.27$ |
| 256 steps | 116.85 | 5.66 | $14.73 \pm 0.19$ | $2'223.63 \pm 28.47$ |
| 512 steps | 111.11 | 5.64 | $28.15 \pm 0.32$ | $1'163.79 \pm 13.25$ |
| 1024 steps | 108.24 | 5.63 | $54.62 \pm 0.66$ | $599.97 \pm 7.27$ |
| *PGM 6 / 6 (dim. 1024, uniform sampling)* | | | | |
| 32 steps | 185.16 | 5.73 | $1.59 \pm 0.01$ | $20'569.99 \pm 95.63$ |
| 64 steps | 138.87 | 5.70 | $3.03 \pm 0.01$ | $10'805.31 \pm 14.11$ |
| 128 steps | 116.95 | 5.67 | $5.93 \pm 0.01$ | $5'518.09 \pm 13.46$ |
| 256 steps | 108.51 | 5.65 | $11.77 \pm 0.01$ | $2'782.78 \pm 3.46$ |
| 512 steps | 101.94 | 5.63 | $23.25 \pm 0.01$ | $1'408.88 \pm 1.05$ |
| 1024 steps | 99.64 | 5.62 | $46.31 \pm 0.02$ | $707.52 \pm 0.34$ |
| *PGM 6 / 6 (dim. 1024, non-uniform sampling)* | | | | |
| 32 steps | 191.30 | 5.74 | $2.12 \pm 0.07$ | $15'415.56 \pm 467.20$ |
| 64 steps | 138.67 | 5.69 | $3.940 \pm 0.06$ | $8'318.72 \pm 135.47$ |
| 128 steps | 118.17 | 5.67 | $7.60 \pm 0.09$ | $4'311.80 \pm 54.92$ |
| 256 steps | 108.93 | 5.65 | $14.84 \pm 0.20$ | $2'207.71 \pm 29.71$ |
| 512 steps | 105.41 | 5.64 | $28.56 \pm 0.33$ | $1'147.17 \pm 13.47$ |
| 1024 steps | 102.93 | 5.62 | $55.50 \pm 0.36$ | $590.37 \pm 3.85$ |

Table 7: Generative perplexity of MDLM and PGM after distillation with varying precision.

| Model | Gen. PPL ↓ | Entropy ↑ | Latency ↓ (ms) | Throughput ↑ (tok/s) |
|---|---|---|---|---|
| *MDLM + SDTT (loss in BF16)* | | | | |
| 32 steps | 66.26 | 5.49 | $8.037 \pm 0.01$ | $4'077.08 \pm 3.06$ |
| 64 steps | 53.98 | 5.46 | $15.82 \pm 0.01$ | $2'070.67 \pm 0.69$ |
| 128 steps | 48.02 | 5.44 | $31.41 \pm 0.01$ | $1'043.22 \pm 0.16$ |
| 256 steps | 45.86 | 5.42 | $62.54 \pm 0.01$ | $523.90 \pm 0.06$ |
| 512 steps | 44.21 | 5.40 | $124.94 \pm 0.16$ | $262.26 \pm 0.33$ |
| 1024 steps | 43.19 | 5.38 | $249.31 \pm 0.11$ | $131.42 \pm 0.05$ |
| *MDLM + SDTT (loss in FP32)* | | | | |
| 32 steps | 61.65 | 5.46 | $8.037 \pm 0.01$ | $4'077.08 \pm 3.06$ |
| 64 steps | 50.65 | 5.43 | $15.82 \pm 0.01$ | $2'070.67 \pm 0.69$ |
| 128 steps | 45.06 | 5.40 | $31.41 \pm 0.01$ | $1'043.22 \pm 0.16$ |
| 256 steps | 41.70 | 5.37 | $62.54 \pm 0.01$ | $523.90 \pm 0.06$ |
| 512 steps | 40.63 | 5.36 | $124.94 \pm 0.16$ | $262.26 \pm 0.33$ |
| 1024 steps | 39.50 | 5.32 | $249.31 \pm 0.11$ | $131.42 \pm 0.05$ |
| *PGM 6 / 6 (dim. 1024) + SDTT (loss in BF16)* | | | | |
| 32 steps | 91.61 | 5.56 | $1.59 \pm 0.01$ | $20'569.99 \pm 95.63$ |
| 64 steps | 72.73 | 5.52 | $3.03 \pm 0.01$ | $10'805.31 \pm 14.11$ |
| 128 steps | 63.83 | 5.49 | $5.93 \pm 0.01$ | $5'518.09 \pm 13.46$ |
| 256 steps | 58.74 | 5.47 | $11.77 \pm 0.01$ | $2'782.78 \pm 3.46$ |
| 512 steps | 58.77 | 5.47 | $23.25 \pm 0.01$ | $1'408.88 \pm 1.05$ |
| 1024 steps | 56.47 | 5.46 | $46.31 \pm 0.02$ | $707.52 \pm 0.34$ |
| *PGM 6 / 6 (dim. 1024) nucleus (p=0.9)+ SDTT (loss in BF16)* | | | | |
| 32 steps | 68.33 | 5.50 | $1.74 \pm 0.01$ | $18'866.12 \pm 18.35$ |
| 64 steps | 53.88 | 5.45 | $3.18 \pm 0.01$ | $10'307.16 \pm 6.58$ |
| 128 steps | 46.99 | 5.42 | $6.10 \pm 0.01$ | $5'375.20 \pm 2.40$ |
| 256 steps | 43.22 | 5.40 | $11.95 \pm 0.01$ | $2'742.74 \pm 1.32$ |
| 512 steps | 42.79 | 5.39 | $23.63 \pm 0.01$ | $1'386.79 \pm 0.69$ |
| 1024 steps | 40.99 | 5.38 | $46.83 \pm 0.02$ | $699.80 \pm 0.24$ |
| *PGM 6 / 6 (dim. 1024) + SDTT (loss in FP32)* | | | | |
| 32 steps | 84.97 | 5.52 | $1.74 \pm 0.01$ | $20'569.99 \pm 95.63$ |
| 64 steps | 67.60 | 5.49 | $3.18 \pm 0.01$ | $10'805.31 \pm 14.11$ |
| 128 steps | 60.06 | 5.47 | $6.10 \pm 0.01$ | $5'518.09 \pm 13.46$ |
| 256 steps | 55.97 | 5.45 | $11.95 \pm 0.01$ | $2'782.78 \pm 3.46$ |
| 512 steps | 54.13 | 5.44 | $23.13 \pm 0.01$ | $1'408.88 \pm 1.05$ |
| 1024 steps | 52.77 | 5.44 | $46.83 \pm 0.02$ | $707.52 \pm 0.34$ |
| *PGM 6 / 6 (dim. 1024) nucleus (p=0.9)+ SDTT (loss in FP32)* | | | | |
| 32 steps | 63.46 | 5.45 | $1.59 \pm 0.01$ | $18'866.12 \pm 18.35$ |
| 64 steps | 49.94 | 5.41 | $3.03 \pm 0.01$ | $10'307.16 \pm 6.58$ |
| 128 steps | 43.84 | 5.39 | $5.93 \pm 0.01$ | $5'375.20 \pm 2.40$ |
| 256 steps | 40.76 | 5.36 | $11.77 \pm 0.01$ | $2'742.74 \pm 1.32$ |
| 512 steps | 39.46 | 5.36 | $23.25 \pm 0.01$ | $1'386.79 \pm 0.69$ |
| 1024 steps | 38.81 | 5.35 | $46.31 \pm 0.02$ | $699.80 \pm 0.24$ |
| *PGM 8 / 8 + SDTT (loss in BF16)* | | | | |
| 32 steps | 102.64 | 5.54 | $1.55 \pm 0.01$ | $21'120.99 \pm 83.59$ |
| 64 steps | 82.93 | 5.50 | $3.00 \pm 0.01$ | $10'914.91 \pm 41.69$ |
| 128 steps | 73.19 | 5.48 | $5.86 \pm 0.02$ | $5'585.57 \pm 24.49$ |
| 256 steps | 70.30 | 5.47 | $11.64 \pm 0.03$ | $2'814.99 \pm 9.33$ |
| 512 steps | 68.07 | 5.46 | $22.98 \pm 0.02$ | $1'425.89 \pm 1.61$ |
| 1024 steps | 65.87 | 5.44 | $45.84 \pm 0.03$ | $714.71 \pm 0.50$ |
| *PGM 8 / 8 + SDTT (loss in FP32)* | | | | |
| 32 steps | 87.64 | 5.51 | $1.55 \pm 0.01$ | $21'120.99 \pm 83.59$ |
| 64 steps | 70.47 | 5.48 | $3.00 \pm 0.01$ | $10'914.91 \pm 41.69$ |
| 128 steps | 62.66 | 5.46 | $5.86 \pm 0.02$ | $5'585.57 \pm 24.49$ |
| 256 steps | 59.38 | 5.45 | $11.64 \pm 0.03$ | $2'814.99 \pm 9.33$ |
| 512 steps | 57.57 | 5.44 | $22.98 \pm 0.02$ | $1'425.89 \pm 1.61$ |
| 1024 steps | 56.12 | 5.44 | $45.84 \pm 0.03$ | $714.71 \pm 0.50$ |

Table 8: Sample quality and efficiency on ImageNet for different numbers of sampling steps using the *Confidence-based* sampler. We generate images in batches of 32 to measure throughput, and use a batch size of 1 to measure latency. Throughput is lower with CFG because each step requires two forward passes (conditional and unconditional). The throughput and latency are averaged over 10 batches.

| Model | FID ↓ | IS ↑ | Latency ↓ (ms) | Throughput ↑ (img/s) |
|---|---|---|---|---|
| *MaskGIT (32 steps; 458M)* | | | | |
| w = 0 | 14.30 | 82.41 | 0.70 | 2.05 |
| w = 1 | 7.80 | 151.62 | 1.21 | 1.05 |
| w = 2 | **6.78** | 208.92 | 1.21 | 1.05 |
| w = 3 | 7.37 | 255.69 | 1.21 | 1.05 |
| w = 4 | 7.46 | 289.93 | 1.21 | 1.05 |
| w = 5 | 9.61 | 250.86 | 1.21 | 1.05 |
| w = 6 | 21.68 | 149.61 | 1.21 | 1.05 |
| *MaskGIT (64 steps; 458M)* | | | | |
| w = 0 | 15.62 | 79.23 | 1.39 | 1.03 |
| w = 1 | 9.40 | 140.48 | 2.41 | 0.52 |
| w = 2 | 8.06 | 195.35 | 2.41 | 0.52 |
| w = 3 | 8.19 | 239.89 | 2.41 | 0.52 |
| w = 4 | **7.61** | 267.26 | 2.41 | 0.52 |
| w = 5 | 11.44 | 202.92 | 2.41 | 0.52 |
| w = 6 | 26.41 | 113.63 | 2.41 | 0.52 |
| *PGM 12 / 12 (32 steps; 464M)* | | | | |
| w = 0 | 18.77 | 67.22 | 1.04 | 6.86 |
| w = 1 | 8.96 | 135.03 | 1.08 | 3.76 |
| w = 2 | **6.67** | 201.66 | 1.08 | 3.76 |
| w = 3 | 7.09 | 255.43 | 1.08 | 3.76 |
| w = 4 | 8.30 | 290.18 | 1.08 | 3.76 |
| w = 5 | 9.59 | 307.52 | 1.08 | 3.76 |
| w = 6 | 10.84 | 313.27 | 1.08 | 3.76 |
| *PGM 12 / 12 (64 steps; 464M)* | | | | |
| w = 0 | 19.45 | 64.44 | 2.04 | 3.52 |
| w = 1 | 10.08 | 124.90 | 2.04 | 1.90 |
| w = 2 | **7.35** | 188.77 | 2.04 | 1.90 |
| w = 3 | 7.39 | 238.31 | 2.04 | 1.90 |
| w = 4 | 8.13 | 276.55 | 2.04 | 1.90 |
| w = 5 | 9.18 | 297.44 | 2.04 | 1.90 |
| w = 6 | 10.38 | 302.29 | 2.04 | 1.90 |
| *PGM 14 / 10 (32 steps; 464M)* | | | | |
| w = 0 | 21.97 | 60.07 | 1.04 | 7.09 |
| w = 1 | 11.24 | 121.39 | 1.04 | 3.90 |
| w = 2 | 8.05 | 183.20 | 1.04 | 3.90 |
| w = 3 | **7.76** | 232.62 | 1.04 | 3.90 |
| w = 4 | 8.47 | 263.73 | 1.04 | 3.90 |
| w = 5 | 9.39 | 288.60 | 1.04 | 3.90 |
| w = 6 | 10.40 | 291.46 | 1.04 | 3.90 |
| *PGM 14 / 10 (64 steps; 464M)* | | | | |
| w = 0 | 22.74 | 56.77 | 2.03 | 3.63 |
| w = 1 | 12.32 | 112.98 | 2.04 | 1.97 |
| w = 2 | 8.80 | 171.11 | 2.04 | 1.97 |
| w = 3 | **8.11** | 219.33 | 2.04 | 1.97 |
| w = 4 | 8.44 | 253.10 | 2.04 | 1.97 |
| w = 5 | 9.12 | 270.16 | 2.04 | 1.97 |
| w = 6 | 9.90 | 279.00 | 2.04 | 1.97 |

Table 9: Sample quality and efficiency on ImageNet for different numbers of sampling steps using the *Halton* sampler. We generate images in batches of 32 to measure throughput, and use a batch size of 1 to measure latency. Throughput is lower with CFG because each step requires two forward passes (conditional and unconditional). The throughput and latency are averaged over 10 batches.

| Model | FID ↓ | IS ↑ | Latency ↓ (ms) | Throughput ↑ (img/s) |
|---|---|---|---|---|
| *MaskGIT (32 steps; 458M)* | | | | |
| w = 0 | 25.72 | 57.70 | 0.70 | 2.02 |
| w = 1 | **5.35** | 267.49 | 1.21 | 1.05 |
| w = 2 | 12.82 | 365.36 | 1.21 | 1.05 |
| w = 3 | 17.24 | **408.30** | 1.21 | 1.05 |
| w = 4 | 15.65 | 365.97 | 1.21 | 1.05 |
| w = 5 | 25.33 | 182.14 | 1.21 | 1.05 |
| w = 6 | 48.97 | 74.74 | 1.21 | 1.05 |
| *MaskGIT (64 steps; 458M)* | | | | |
| w = 0 | 18.61 | 69.68 | 1.39 | 1.03 |
| w = 1 | **6.76** | 283.96 | 2.41 | 0.52 |
| w = 2 | 14.97 | 372.74 | 2.41 | 0.52 |
| w = 3 | 18.30 | **410.60** | 2.41 | 0.52 |
| w = 4 | 16.18 | 312.75 | 2.41 | 0.52 |
| w = 5 | 32.69 | 126.69 | 2.41 | 0.52 |
| w = 6 | 60.12 | 51.40 | 2.41 | 0.52 |
| *PGM 12 / 12 (32 steps; 464M)* | | | | |
| w = 0 | 22.59 | 66.32 | 1.03 | 13.44 |
| w = 1 | 10.21 | 134.22 | 1.03 | 7.93 |
| w = 2 | 6.04 | 203.28 | 1.03 | 7.93 |
| w = 3 | **5.54** | 263.53 | 1.03 | 7.93 |
| w = 4 | 6.38 | 311.53 | 1.03 | 7.93 |
| w = 5 | 7.58 | 345.18 | 1.03 | 7.93 |
| w = 6 | 8.83 | **372.07** | 1.03 | 7.93 |
| *PGM 12 / 12 (64 steps; 464M)* | | | | |
| w = 0 | 16.19 | 79.26 | 1.96 | 7.17 |
| w = 1 | 6.67 | 151.44 | 2.01 | 4.12 |
| w = 2 | **4.56** | 218.98 | 2.01 | 4.12 |
| w = 3 | 4.98 | 276.16 | 2.01 | 4.12 |
| w = 4 | 6.47 | 322.53 | 2.01 | 4.12 |
| w = 5 | 7.95 | 352.56 | 2.01 | 4.12 |
| w = 6 | 9.47 | **379.39** | 2.01 | 4.12 |
| *PGM 14 / 10 (32 steps; 464M)* | | | | |
| w = 0 | 25.42 | 62.11 | 1.01 | 12.70 |
| w = 1 | 11.57 | 128.22 | 1.00 | 7.37 |
| w = 2 | 6.60 | 196.56 | 1.00 | 7.37 |
| w = 3 | **5.55** | 253.76 | 1.00 | 7.37 |
| w = 4 | 6.00 | 301.66 | 1.00 | 7.37 |
| w = 5 | 7.04 | 334.29 | 1.00 | 7.37 |
| w = 6 | 8.16 | **365.56** | 1.00 | 7.37 |
| *PGM 14 / 10 (64 steps; 464M)* | | | | |
| w = 0 | 18.03 | 75.40 | 1.93 | 6.69 |
| w = 1 | 7.77 | 144.26 | 1.99 | 3.80 |
| w = 2 | **4.76** | 212.39 | 1.99 | 3.80 |
| w = 3 | 4.85 | 268.55 | 1.99 | 3.80 |
| w = 4 | 5.88 | 309.88 | 1.99 | 3.80 |
| w = 5 | 7.23 | 342.40 | 1.99 | 3.80 |
| w = 6 | 8.63 | **366.57** | 1.99 | 3.80 |

Table 10: Throughput (TP) of MDLM and PGM with a context length of 4096, for varying number of inference steps. PGM is significantly faster than MDLM.

| Model | TP (4096) | TP (1024) | TP (256) | TP (64) |
|---|---|---|---|---|
| MDLM | $30.45 \pm 0.06$ | $121.25 \pm 0.02$ | $483.53 \pm 0.25$ | $1'912.16 \pm 1.44$ |
| PGM 8/8 | $128.99 \pm 0.23$ | $697.36 \pm 32.83$ | $\mathbf{2'216.91} \pm 3.06$ | $\mathbf{8'203.82} \pm 6.60$ |
| PGM 6/6 (dim=1024) | $\mathbf{129.01} \pm 0.67$ | $\mathbf{706.65} \pm 36.23$ | $2'146.60 \pm 15.12$ | $8'175.69 \pm 7.85$ |

