# OpenReview forum: "Partition Generative Modeling: Masked Modeling Without Masks"
_ICLR.cc/2026/Conference — ICLR 2026 Oral_

### Official Review · Reviewer_m818 · 2025-10-19

**Soundness:** 4
**Presentation:** 3
**Contribution:** 4
**Rating:** 8
**Confidence:** 3

**Summary:**

This paper introduces the PGM, a new framework with core architecture innovation that combines the strengths of AR and MGM. PGM partitions tokens into two disjoint groups and constrains attention such that each group predicts the other. This removes the need for explicit MASK tokens while preserving parallel decoding and arbitrary generation order and address the training inefficiency of MGM.
Experiments on LM1B, OpenWebText, and ImageNet show that PGMs achieve up to 5–7× faster inference throughput than MDLM and MaskGIT, with similar or better metrics including perplexity and FID. PGMs also support distillation for additional speedups.
This work is overall sound to me, but I am not an expert in architecture design.

**Strengths:**

1. Good novelty: replacing masking with partitioning is a simple yet powerful idea that effectively unifies the efficiency of AR models with the flexibility of MGMs.
2. Solid architectural design: The GroupSwap mechanism and partition-wise attention are well-motivated and carefully engineered to achieve the partition mechanism.
3. The experimental results are strong, with improved performance for both text and image generation.

**Weaknesses:**

1. While the PGM is motivated be the inefficiency of MDM training, the authors are encouraged to provide evidence to show faster learning/convergence of PGM than MDLM. This probably relates to the training stability.

**Questions:**

1. Does "PGM 8 / 8"  mean 8 layers of encoder and 8 layers of decoder?
2. The origin of the training instability. Do authors still observe this when PGM is trained without complementary masking.

---

> ### Author Response · Authors · 2025-11-21
>
> We are grateful to the reviewer for the time devoted to our submission. We respond to the questions below.
>
> ### Question: convergence speed compared to MDLM
>
> - In section 5.3, we train an MDLM with 2x larger batches, each containing 2 copies of each training sequence, complementarily masked (i.e., if a token is masked in one, it is *not* masked in the other). **This improves the validation perplexity on both LM1B (27.67 → 26.80) and OpenWebText (23.07 → 22.98)** after 1M steps. We compare the wall-clock training speed and throughput of PGM and MDLM in Appendix D.
>
> ### Question: notation (PGM 8 / 8)
>
> - “PGM 8 / 8” in Table 1 indeed represents a PGM with 8 encoder and decoder layers (caption of Table 1, lines `284`).
>
> ### Question: training stability
>
> - *Complementary masking* produces occasional loss spikes for both MDLM and PGM (Fig. 5), but these are relatively rare over 1M steps. Importantly, **they don’t cause divergence and do not require special handling**.
> - We did not train a PGM variant without complementary masking, because applying loss to all positions is a key advantage of PGM over MGMs and, as shown in Section 5.3, can improve performance.

---

> > ### Comment · Reviewer_m818 · 2025-11-27
> >
> > Thank you for the detailed response and my concerns have been addressed. The score remains the same.

---

> > > ### Author Response · Authors · 2025-11-27
> > >
> > > We are glad that our revisions have addressed your concerns, and we appreciate your positive assessment of our work.

---

### Official Review · Reviewer_DktD · 2025-10-27

**Soundness:** 3
**Presentation:** 4
**Contribution:** 3
**Rating:** 8
**Confidence:** 4

**Summary:**

This paper proposes avoiding the repetitive computation of the `MASK` token in masked generative modeling (MGM), switching from the decoder-only MGM to an encoder-decoder architecture. The inference model is defined as:
1. self-attention **only within known indices**.
2. cross-attention swapping to unknown indices (with opposite-group masking to prevent leakage).
3. cross-attention **only within unknown indices** (projecting to the embeddings of stage one).

The complexity is still O(L^2) (L = sequence length) but with a significantly smaller coefficient (encoder $(L-k)^2$ + decoder $k(L−k)$ per step when remaining sequence length = k).
The practical speedup is about 5x and is scalable, with comparable generation quality against MGM.
Distillation-accelerated models maintains the acceleration against MGM.

**Strengths:**

- The empirical benefit is strong: 5x faster than MGM (4.6x faster with nucleus sampling).

- Complementary masking is a smart and original trick to let one training step effectively count as two steps.

- Section 5.3: fair comparison against MDLM (MGM) by isolating the complementary masking trick.

- The down-stream tasks spreads across image and language, and the evaluation is solid. Distillation is also explored, which improves the practical significance of the paper.

**Weaknesses:**

- The fairness of Table 2's comparison is not immediately visible—I believe the fairness should outweigh matching performance. Since the paper switches from decoder-only to encoder-decoder architecture, controlling hyperparameters (width, head, depth and MLP width multipliers) seems crucial to get a fair comparison.
In LM1B, it is a good idea controlling parameter counts and comparing with PGM(6/6)\~170M, but in OWT, that model is missing in the main text (only the dim. 1024 model is shown). I don't understand why it only appears in the appendix.

- I don't understand the labels (5.3) (5.4) (5.5) in Figure 4 (right).

- Minor: "sparse attention" is used to describe the masking mechanism, but I believe it is an overuse of the term, as the mask is not actually sparse—perhaps group-wise attention is more suitable.

**Questions:**

- Except for the top-k/nucleus confident tokens, the computations are wasted. I wonder if it is possible to reuse these noisy states instead of re-initializing decoder queries at each denoising step?

- The current decoder architecture is cross-attention-only—which makes it easy to control parameter count c.f. MDLM, but lacks the standard self-attention component. Have you thought about this variant?

- The information exchange from known to unknown indices entirely relies on the swap xattention layer. I wonder if it is possible to do the exchange in each decoder layer instead? (Of course this will make the complimentary masking trick not possible.)

---

> ### Author Response · Authors · 2025-11-21
>
> We thank the reviewer for the effort invested in reviewing our paper. We appreciate the questions and respond below.
>
> ### Question: comparison with the PGM 6 / 6 variant (OWT) in Table 2
>
> - In Tables 1 and 2, we compare models trained for 1M steps. Since the PGM 6/6 variant was weaker after 200k steps, we did not train it further to reduce its costs and reported the 200k-step results in the appendix.
>
> ### Question: architecture sweeps
>
> - Due to limitations in our compute budget, we prioritized the complementary masking experiments (section 5.3) because we believe they could provide more general insights.
>
> ### Question: text labels in Figure 4
>
> - The labels represent the average unigram entropy of the generated text as a proxy for diversity [1]. We further emphasized in the updated manuscript (lines `341-343`).
>
> ### Concern: sparse attention
>
> - We replaced “sparse attention” with “group-wise attention” in the updated manuscript, following your suggestion.
>
> ### Question: Reuse noisy states to initialize the decoder queries
>
> - The main challenge with this decoder query initialization is determining how to train the model to process noisy states during sampling, since noisy query initializations cannot be obtained in a single forward pass during training. Therefore, we left it to future work.
>
> ### Question: self-attention in the decoder
>
> - Early on, we tried using self-attention in the decoder (alternating self and cross attention), but after 200k steps, it did not improve over the cross-attention-only design. We therefore stopped training to save costs.
>
> ### Question: information exchange and GroupSwap layer
>
> - Information is *first* exchanged across positions through the GroupSwap layer, but not *exclusively*. Indeed, the decoder attends to the encoder's output, so information does not need to be exchanged in each layers.
>
> [1] Continuous diffusion for categorical data, Dieleman et al., 2022

---

> > ### Comment · Reviewer_DktD · 2025-11-27
> >
> > Thanks for the precise and effective replies; the execution of the idea is solid. I would like to maintain my rating.

---

> > > ### Author Response · Authors · 2025-11-27
> > >
> > > We are glad our responses were helpful, thank you for your positive evaluation of our work

---

### Official Review · Reviewer_pTqP · 2025-10-31

**Soundness:** 3
**Presentation:** 3
**Contribution:** 3
**Rating:** 6
**Confidence:** 2

**Summary:**

This paper introduces a new generative modeling framework for language modeling, termed Partition Generative Models (PGM), aimed at improving inference efficiency. Unlike Masked Generative Models (MGM), PGM avoids applying the forward process to masked tokens, thereby reducing computational cost. The authors present tailored architectural modifications, along with corresponding training and inference strategies, to enable efficient generation within this framework. Experimental results indicate that PGM achieves faster inference than existing MGM approaches while maintaining comparable generation quality.

**Strengths:**

1. The core idea of avoiding computation on masked tokens during inference, along with the corresponding training strategy, is interesting and effectively targets a key inefficiency in existing masked generative models.

2. The empirical results demonstrate that PGM can significantly accelerate inference while maintaining generation quality comparable to other state-of-the-art generative models, supporting the practical value of the proposed approach.

3. The paper is clearly written, well-structured, and easy to follow, making the technical contributions accessible to the reader.

**Weaknesses:**

I did not identify any major weaknesses in this paper. I do, however, have one question for clarification:

The proposed training pipeline includes two prediction components that operate on the same batch of data, which suggests that training efficiency could potentially be better than MDLM. Could the authors provide quantitative results or analysis regarding training efficiency, such as training speed, computational cost, or resource usage compared to MDLM?

**Questions:**

NA

---

> ### Author Response · Authors · 2025-11-21
>
> We are grateful to the reviewer for their assessment of our work and for the comments provided. Find our responses below.
>
> ### Efficiency of complementary predictions in the same sequence
>
> - In section 5.3, we train an MDLM with 2x larger batches, each containing two copies of each training sequence, complementarily masked (i.e., if a token is masked in one, it is *not* in the other). **This can improve the validation perplexity on both LM1B (27.67 → 26.80) and OpenWebText (23.07 → 22.98).**
>
> ### Computational costs
>
> - **We report the throughput and total training durations in Appendix D.** The current implementation trains slower than MDLM, because we use `torch.sdpa` since flash-attention does not support arbitrary attention masks.
> - However, *PGMs use highly structured attention patterns*. Indeed, partitioning can be implemented by shuffling tokens and choosing a random split point for the two groups, resulting in block-diagonal attention.
> - Future work will focus on custom attention kernels that leverage the block-diagonal structure, which *could train faster than MDLM, much as causal attention is cheaper than full bidirectional attention* (*“make it work, make it right, make it fast”).*

---

> > ### Comment · Reviewer_pTqP · 2025-11-27
> >
> > Thanks for the explanation. I will keep my positive rating.

---

### Official Review · Reviewer_YDst · 2025-11-01

**Soundness:** 3
**Presentation:** 3
**Contribution:** 3
**Rating:** 6
**Confidence:** 3

**Summary:**

The authors introduce Partition Generative Models (PGMs), based on the observation that masked generative models (MGMs) waste compute on masked tokens, which contain no information.
Instead of masking tokens, PGMs partition the input tokens into two disjoint groups and train the model to predict one group from the other.
This approach allows the model to process only unmasked tokens which eliminating the need for explicit masking and leads to significantly faster sampling.

**Strengths:**

- The GroupSwap layer and partition-aware transformer structure are well-motivated
- Includes analyses of perplexity, latency, throughput, and ablations on masking vs. partitioning.
- Strong empirical results across both text and image generation tasks, PGMs deliver substantial inference speedups (up to 7×) with little to no degradation in output quality.

**Weaknesses:**

- The architectural details (e.g., data-dependent vs. data-independent queries) are dense and could be clarified or simplified, the paper is a bit difficult to follow.
- The largest experiments are modest in size (268M parameters). It remains unclear if PGMs scale favorably compared to state-of-the-art large AR or diffusion model
- No comparison against recent SOTA model non-autoregressive language models beyond MDLM.

**Questions:**

- How does the choice of partition ratio (t) affect convergence and quality? Is it dynamically sampled or fixed?
- why cant you use KVcache that would reduce the time complexity from sampling in MGM? Would PGM will be still faster?

---

> ### Author Response · Authors · 2025-11-21
>
> We thank the reviewer for taking the time to evaluate our submission and appreciate the feedback. We address the questions below.
>
> > The architectural details (e.g., data-dependent vs. data-independent queries) are dense
> >
> - We emphasize the design choices around “GroupSwap,” as they are crucial to understanding why PGMs can train without leaking targets. We tightened section 4.1 (description of the GroupSwap layer) following your comment.
>
> > It remains unclear if PGMs scale favorably compared to state-of-the-art large AR or diffusion model
> >
> - Our models have up to ~260M parameters for text and ~460M for ImageNet, matching the MaskGIT baseline [2]. Training larger models was too expensive, hence we could not scale PGMs further. We simply show PGMs can sample 5-5x faster on text and up to 7x faster on ImageNet256 at the current size.
>
> > Comparison against recent SOTA model non-autoregressive language models
> >
> - Semi-AR models [3] and any-order autoregressive models [4] are discussed in the related work. We focus on MGMs such as MDLM [1] and MaskGIT [2] **because our goal is to speed them up without extra assumptions**, whereas [3-4] differ in key ways from MGMs. Specifically,
>     - [3] generates tokens block-by-block, left-to-right, so it cannot generate in any order. *It also requires two forward passes per training step* (MGMs/PGMs need only one) and optimizes a different ELBO than MDLM [1].
>     - AO-ARMs [4] use causal transformers, so they lack bidirectional attention between clean tokens, unlike MGMs. They also rely on heuristics to avoid training on an exponential number of sequence orderings, which wastes model capacity.
> - In contrast, PGMs generate tokens in any order, allow bidirectional attention between clean tokens, and optimize a simple objective equivalent to the MGM objective.
>
> > Is [the partitioning ratio] dynamically sampled or fixed?
> >
> - **We use a partitioning ratio that matches MDLM’s masking ratios**. If MDLM masks a fraction $c$ of the tokens, PGM assigns a fraction $c$ to group 0. This is described on lines `242-246` in the original manuscript, and further highlighted on lines `245-246` in the revised version.
>
> > Why cant you use KV cache that would reduce the time complexity from sampling in MGM?
> >
> - ***Exact* KV caching is not compatible with MGMs and PGMS** because they use bidirectional attention. However, recent work [5-6] proposed *approximate* caching by freezing certain activations during sampling, which trades off quality for speed. Therefore, **approximate caching is orthogonal to our contributions** and can be applied to both MGMs and PGMs.
>
> [1] Simple and Effective Masked Diffusion Language Models, Sahoo et al., 2024
>
> [2] Halton Scheduler For Masked Generative Image Transformer, Besnier et al., 2025
>
> [3] Block Diffusion: Interpolating Between Autoregressive and Diffusion Language Models, Arriola et al., 2025
>
> [4] Training and Inference on Any-Order Autoregressive Models the Right Way, Shih et al., 2022
>
> [5] dKV-Cache: The Cache for Diffusion Language Models, Ma et al., 2025
>
> [6] Fast-dLLM: Training-free Acceleration of Diffusion LLM by Enabling KV Cache and Parallel Decoding, Wu et al., 2025.

---

### Author Response · Authors · 2025-12-03
**General Response**

We sincerely thank all reviewers for their thoughtful feedback. We are grateful to the ACs for their work as well. As the discussion period closes, we summarize our main contributions.

---

## Rethinking Masked Generative Modeling

Masked generative models (MGMs), especially diffusion language models [1-3], have recently become competitive with autoregressive models for language modeling, but **they remain inefficient because they process many uninformative masked tokens during sampling**. All reviewers rated the soundness, presentation, and contributions of our work as good or excellent, and **there is broad agreement that PGM provides a simple, novel, and practical direction for efficient generative modeling**.

## Our contributions

**Partition Generative Models (PGMs) significantly improves the throughput of MGMs** by partitioning tokens instead of masking them, and adopting a transformer architecture that enforces information separation across groups. Our main contributions are threefold:

1. **A simple and effective modeling framework.** PGMs eliminate `MASK` tokens entirely through token partitioning. Importantly, **PGMs require no additional assumptions beyond those of MGMs** and use the same training objective, making the approach directly applicable across modalities, such as text and images.
2. **Inference speedups.** PGMs achieve at least 5x higher throughput than standard diffusion language models [1-3] on OpenWebText while improving generative perplexity. **PGMs outperform GPT-2 (using KV-caching) in Generative Perplexity**, when matching their wall-clock throughputs (Fig. 1, page 1), even without distillation. On ImageNet, PGMs deliver up to 7x higher throughput than MaskGIT [4] with competitive FID.
3. **Compatibility with distillation and guidance.** Owing to the simplicity of PGMs, many effective techniques developed for MGMs transfer with minimal effort. In particular, **PGMs remain compatible with existing distillation methods** [5]**, classifier-free guidance** [6]**, and alternative sampling procedures** [7]**.**

---

[1] Shi et al., 2024 “Simplified and Generalized Masked Diffusion for Discrete Data”

[2] Sahoo et al., 2024 “Simple and Effective Masked Diffusion Language Models”

[3] Ou et al., 2024 “Your Absorbing Discrete Diffusion Secretly Models the Conditional Distributions of Clean Data”

[4] Chang et al., 2022 “MaskGIT: Masked Generative Image Transformer”

[5] Deschenaux and Gulcehre, 2025 “Beyond Autoregression: Fast LLMs via Self-Distillation Through Time”

[6] Ho et al., 2022 “Classifier-Free Diffusion Guidance”

[7] Besnier et al., 2025 “Halton Scheduler For Masked Generative Image Transformer”

---

### Meta-Review · Area_Chair_r4jU · 2026-01-18

**Summary:**

This paper proposes Partition Generative Models. It removes explicit MASK tokens in masked generative modeling by partitioning tokens into two groups, and using group-wise attention so each group predicts the other. This allows the model to avoid processing uninformative masked tokens during sampling, brining large efficiency gains while maintaining performance. Empirically, PGMs report big sampling throughput improvements on various image/text datasets with only small quality changes.

All reviewers reach a consensus that the idea is simple, novel, and practical, with strong empirical results. Overall, I recommend Accept.

**Reviewer Concerns:**

Most reviewer concerns were addressed in the rebuttal.

**Reviewer Scores:**

The reviewers will likely keep the same positive scores.

---

### Decision · Program_Chairs · 2026-01-26

Accept (Oral)